# Eco-HAB as a fully automated and ecologically relevant assessment of social impairments in mouse models of autism

**Alicja Puścian[1], Szymon Łęski[1], Grzegorz Kasprowicz[2,3], Maciej Winiarski[1], Joanna Borowska[1], Tomasz Nikolaev[1], Paweł M Boguszewski[1], Hans-Peter Lipp[4,5], Ewelina Knapska[1]***

[1]Department of Neurophysiology, Nencki Institute of Experimental Biology of Polish Academy of Sciences, Warsaw, Poland; [2]Center for Theoretical Physics, Polish Academy of Sciences, Warsaw, Poland; [3]Institute of Electronic Systems, Warsaw University of Technology, Warsaw, Poland; [4]Institute of Anatomy, University of Zurich, Zurich, Switzerland; [5]School of Laboratory Medicine, Kwazulu-Natal University Durban, Durban, Republic of South Africa

**Abstract** Eco-HAB is an open source, RFID-based system for automated measurement and analysis of social preference and in-cohort sociability in mice. The system closely follows murine ethology. It requires no contact between a human experimenter and tested animals, overcoming the confounding factors that lead to irreproducible assessment of murine social behavior between laboratories. In Eco-HAB, group-housed animals live in a spacious, four-compartment apparatus with shadowed areas and narrow tunnels, resembling natural burrows. Eco-HAB allows for assessment of the tendency of mice to voluntarily spend time together in ethologically relevant mouse group sizes. Custom-made software for automated tracking, data extraction, and analysis enables quick evaluation of social impairments. The developed protocols and standardized behavioral measures demonstrate high replicability. Unlike classic three-chambered sociability tests, Eco-HAB provides measurements of spontaneous, ecologically relevant social behaviors in group-housed animals. Results are obtained faster, with less manpower, and without confounding factors.

***For correspondence:**
e.knapska@nencki.gov.pl

**Competing interests:** The authors declare that no competing interests exist.

## Introduction

Social interactions are complex on any number of levels, from the behavior of individuals to the induced patterns of neuronal activation. They are very difficult to study because even small changes in experimental conditions can produce significant modifications of the behavioral outcome. Experiments must be designed to ensure control over factors affecting these interactions.

Conventional tests of social phenotyping have repeatedly proven inefficient in differentiating certain genotypes and replicating these differences across laboratories and protocol conditions (*Chadman et al., 2008*; *Moy et al., 2004*; *Pearson et al., 2010*; *Tabuchi et al., 2007*). For example, in most studies of autism and sociability in mice, behavioral effects related to anxiety and susceptibility to stress have been overlooked, even though, in humans, both these factors are co-morbid to autism spectrum disorders (*Simonoff et al., 2008*). In fact, anxiety is the most common cause of social impairments in humans. Since some people with ASD never develop so-called 'associated' anxiety, we can assume that underlying neural mechanisms are at least partially different. Therefore, it is important to model single symptoms, separating them to the largest possible extent from any confounding factors in order to understand the brain pathology underlying the observed effects.

Reliable behavioral tests allowing for differential diagnosis are the first necessary step on the long path to disentangling the complex neural background of specific pathologies.

Social interactions of rodents are often assessed using the three-chambered apparatus social approach test (3ChA), in which lone mice are given the choice between approaching a caged conspecific or an inanimate object. The popularity of 3ChA stems from its simplicity, inexpensive cage construction, and the lack of alternative tests. However, its poor cross-laboratory standardization and reproducibility call for alternative testing. At times, different laboratories have come to opposite conclusions concerning sociability of the same autistic phenotype mouse model (*Chadman et al., 2008*; *Tabuchi et al., 2007*, p. 3). Even within a single laboratory, reproducible results can be difficult to obtain (*Jamain et al., 2008*; *El-Kordi et al., 2013*).

Irreproducibility of conventional behavioral tests across laboratories (*Crabbe et al., 1999*) has recently been identified as one of the most important threats to science and its public understanding (*Nature, 2015*; *Morrison, 2014*). The known factors that lead to severe behavioral abnormalities in both males and females (*Beery and Kaufer, 2015*; *Heinrichs and Koob, 2006*; *Sandi and Haller, 2015*) and which are particularly difficult to control include: handling by human experimenters with unique scents and variable handling abilities (*Chesler et al., 2002*; *Sorge et al., 2014*), levels of animal familiarity with the experimental environment, and housing of animals in social isolation (a practice forbidden by EU ethical standards unless justified by experimental requirements). These confounding factors, all present in published 3ChA results (*El-Kordi et al., 2013*), can be eliminated by development of automated, ethologically relevant behavioral tests, which measure spontaneous sociability in group-housed, familiar mice without the presence of a human experimenter.

Irreproducibility and high manpower costs of manual testing have led to the development of automated behavioral tests. These allow assessment of individual behavior of group-housed rodents using either radio-frequency-based identification (RFID) (*Galsworthy et al., 2005*; *Knapska et al., 2006*; *Voikar et al., 2010*; *Schaefer and Claridge-Chang, 2012*; *Howerton et al., 2012*) or advanced video/image processing (*de Chaumont et al., 2012*; *Pérez-Escudero et al., 2014*; *Shemesh et al., 2013*; *Weissbrod et al., 2013*). Video-based systems, often employed for tracking social interactions, have serious limitations in ethologically relevant settings containing shadowed areas and narrow corridors. Scientists try to overcome those difficulties i.a. by combining existing systems with additional tracking methods (*Weissbrod et al., 2013*). Although RFID systems are more invasive than video-tracking solutions due to the necessity of injecting animals with electronic tags, high intra- and inter-laboratory reliability have been confirmed (*Codita et al., 2012*; *Krackow et al., 2010*; *Puścian et al., 2014*). However, their present commercial form is not suitable for measuring sociability.

To meet these challenges, we designed Eco-HAB. This is a fully automated, open-source system based on RFID technology and inspired by the results of ethological field studies in mice (*Dell'omo et al., 1998*, *2000*; *Lopucki and Szymroszczyk, 2003*; *Andrzejewski, 2002*; *Lewejohann et al., 2004*; *Lopucki, 2007*; *Daan et al., 2011*; *von Merten et al., 2014*; *Chalfin et al., 2014*). Group-housed animals equipped with RFID tags live in a spacious, four-compartment apparatus with shadowed areas and narrow tunnels resembling natural burrows. Eco-HAB reduces stress by tracking the tendency of animals to voluntarily spend time together in an environment to which they have already been accustomed and utilizes novel sociability measures for group-housed mice. The system is equipped with software for automated data extraction and analysis, enabling quick evaluation of social activity.

By comparing Eco-HAB results from several mouse models having different sociability levels, we show that this apparatus provides results comparable to the classic three-chamber test when carried out in stress-reducing conditions for single-housed animals. As a result of the innovative electronic solutions (*Figure 1—figure supplement 1*) developed for Eco-HAB, data from this system are obtained much faster, with high reliability (*Figure 1—figure supplement 2*), less manpower (*Figure 1—figure supplement 3*), and are not confounded by the factors that usually blur results of manual testing. The cost of building an Eco-HAB system, suitable for testing up to 12 animals at the same time (approximately 2000 EUR), is comparable to the cost of one three-chambered apparatus. To illustrate the need for automated testing of social behaviors, we also demonstrate how easily one can obtain apparently opposite conclusions regarding sociability of tested mice when confounding factors are not controlled.

## Results and discussion

### Eco-HAB – ethologically relevant testing of social behaviors

The Eco-HAB system and its testing protocols take into account the innate murine tendency to avoid open areas and inhabit enclosed spaces, from which they regularly explore large territories, mostly at night (*Andrzejewski, 2002*; *Dell'Omo et al., 2000*, *1998*). Eco-HAB (*Figure 1* and *Video 1*) consists of four housing compartments, occupying four corners of a larger square, bridged by tube-shaped corridors. These corridors enable mice to travel freely and select preferred areas within the

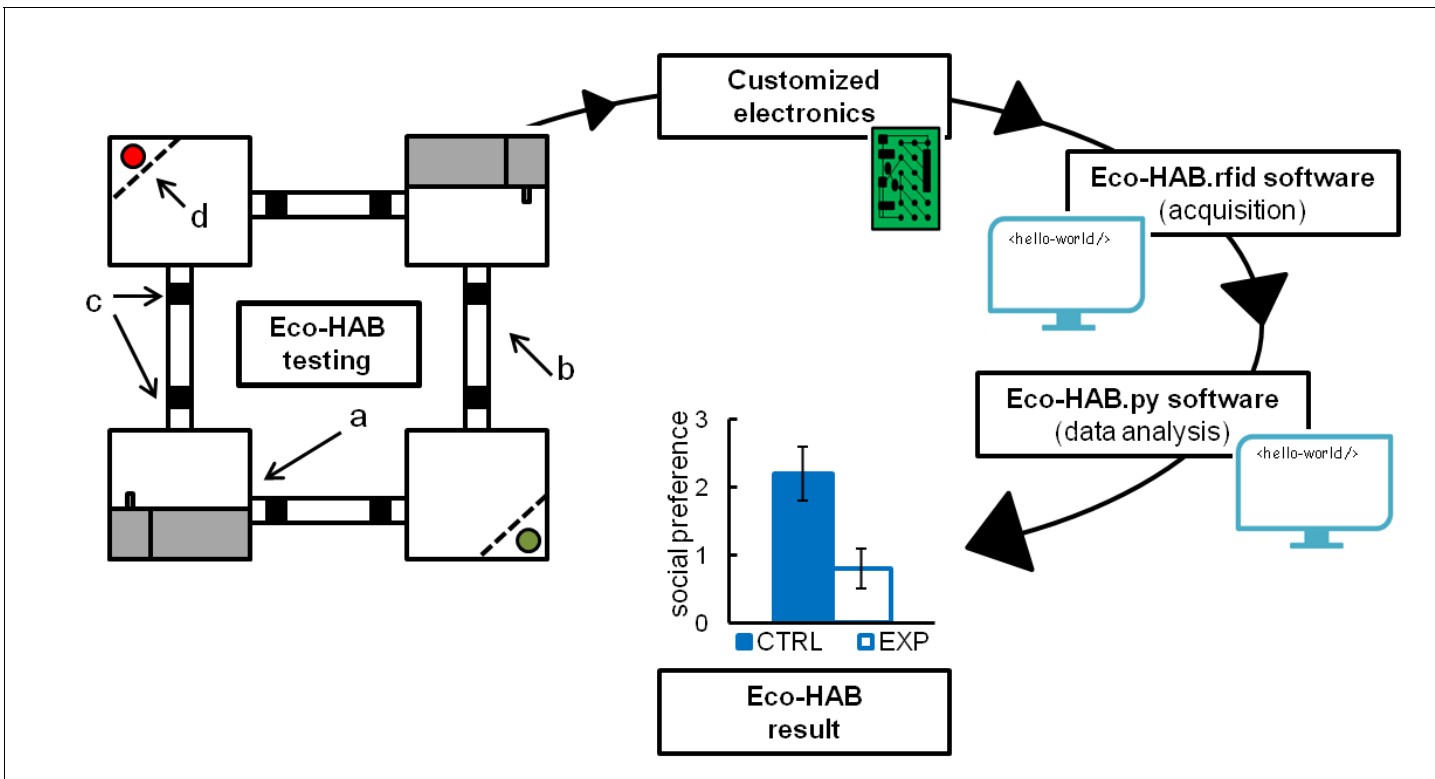

**Figure 1.** A schematic representation of Eco-HAB system and data processing. Eco-HAB consists of four housing compartments (**a**), tube-shaped inter-territorial passages (**b**), radio-frequency identification antennas (**c**), and impassable, perforated partitions behind which social and non-social (control) provenance stimuli may be presented (**d**, red/green dots). Food and water is available in housing compartments adjacent to those containing partitions. Eco-HAB is equipped with customized electronics and two software packages: Eco-HAB.rfid (for data acquisition and collection) and Eco-HAB.py (for filtering corrupted data segments and performing tailored analysis). For a detailed system and software description, see 'Materials and methods'.

The following figure supplements are available for figure 1:

**Figure supplement 1.** Block schematic diagram of customized electronic system for Eco-HAB.

**Figure supplement 2.** RFID antenna efficiency compared to video-based manual scoring.

**Figure supplement 3.** Comparison of time (person-hours) needed for Eco-HAB testing versus three-chambered apparatus testing (stress reducing conditions) of a group of 12 mice.

**Figure supplement 4.** Eco-HAB measures in-cohort sociability in mice.

**Figure supplement 5.** Eco-HAB allows for a detailed analysis of subjects' preference to spend time with another mouse from a tested cohort.

**Figure supplement 6.** Monitoring of subjects' dispersal within Eco-HAB territory for exemplary cohorts of (**a**) C57BL/6 and (**b**) BALB/c mice.

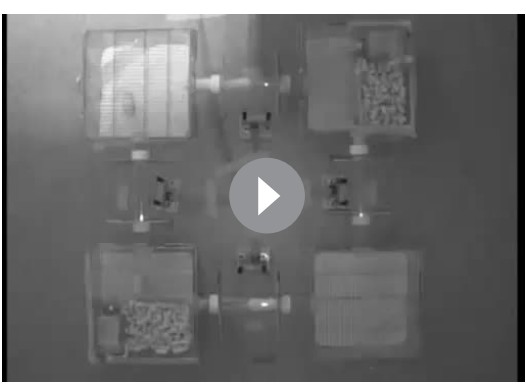

**Video 1.** Top view of the working Eco-HAB. Flashing lights indicate activation of RFID antennas – sensors of the individual recognition system. The clip presents a 30 s period at the beginning of the adaptation phase, when animals are eagerly exploring new territory.

available territory. Two chambers on opposing corners of the square offer access to food and water (*ad libitum*) and provide shelter and secluded places where mice can sleep and rest. The two other compartments have similar designs except that they contain no food or water and one of the corners is equipped with an impassable, transparent, and perforated partition behind which an olfactory stimulus may be presented. Acknowledging the natural tendency of mice to live in family-based groups having many members (*Andrzejewski, 2002*), Eco-HAB is designed for testing littermate cohorts of up to 12 subjects. Animals are tracked individually by subcutaneously injected microtransponders that emit a unique identification code when mice pass under RFID antennas placed on both ends of each corridor. As Eco-HAB is a computer-controlled system, it eliminates human handling and allows for continuous data collection lasting for days or even weeks with minimal presence of

human observers. Since the position of every mouse in an Eco-HAB system can be tracked, a novel in-cohort measure of sociability, based on the tendency of mice for spending time together, can be assessed. The in-cohort sociability score is calculated as described in the Materials and methods and *Figure 1—figure supplements 4–6*. Notably, results show that both Eco-HAB measures–in-cohort sociability and scent-based social approach–allow similar conclusions about mouse behavior to be reached. The latter measure is equivalent to the most natural social exploratory behavior observed in wild populations of mice.

Odor-mediated communication is crucial for survival and plays a key role in all murine social behaviors: mating and reproduction, territory maintenance, development of stable inter-group hierarchy (*Stockley et al., 2013*), and integration of populations of mice in the wild (*Andrzejewski, 2002*). Mice have developed the ability to learn and remember information associated with olfactory cues as effectively as primates recall visually related cues (*Schellinck et al., 2008*). It has been shown that unfamiliar rodents in their natural habitats tend to avoid each other and, if forced to interact openly, often become aggressive (*Lopucki, 2007*). Unfamiliar mice, irrespective of their sexes, are attracted by the scent of a conspecific rather than by its presence (*Andrzejewski, 2002*; *Lopucki and Szymroszczyk, 2003*). For that reason, scents have been previously employed in the 3ChA test (*Ryan et al., 2008*) although, more commonly, unfamiliar animals are introduced into the social chamber. Even though the latter can be implemented in Eco-HAB, in the following experiments we used olfactory stimuli as a more ecologically pertinent solution. Presentation of odors behind partitions prevents spreading of scented bedding over the whole territory, but allows mice to freely approach olfactory cues (for a detailed apparatus and applied electronics description see 'Materials and methods' and *Figure 1* and its *Figure 1—figure supplement 1*). We optimized the behavioral protocol with respect to different testing times and measures. We optimized the behavioral protocol, testing times, and measures to fit with mouse preference. Under these optimized conditions, replicable results were obtained. In the final protocol, cohorts of 7 to 12 same-sex mice are subjected to 72-hr testing. During an adaptation phase (first 48 hr) mice can freely explore the whole apparatus. The odor-based social-preference testing phase starts with simultaneous introduction of two different beddings to two testing compartments. One of the beddings comes from a cage housing a mouse of the same sex, age, and strain as the tested animals ('social' scent), while the other is plain, new bedding from stock. These beddings are placed behind the perforated partitions of the testing compartments (for more details, see the 'Materials and methods'). Mice are allowed to explore both stimuli for 24 hr. Social approach is measured as the relative increase in time spent in the compartment containing social scent divided by the time spent in the opposite chamber that contains bedding without the social scent.

## Experimental stress interferes with results of manual tests of sociability

To illustrate the influence of typical confounding factors (listed in *Figure 2A*) affecting 3ChA social approach testing, we compared the results of 3ChA tests performed in stress-reducing (low stress) and conventional laboratory conditions (high stress). Experiments were performed on two widely used strains of mice displaying different anxiety levels: C57BL/6 and BALB/c. To obtain low-stress conditions, we used mild lighting and extensively habituated both the subjects and mice used as social stimuli to an experimenter and experimental rooms (for a detailed description of the protocol, see 'Materials and methods'). The results of these tests are shown in *Figure 2B* through *Figure 2E*. BALB/c mice (*Figure 2B*) showed social preference only in low-stress conditions (n = 17), and they avoided social interactions when tested in a typical experimental setting (n = 11). In contrast, C57BL/6 mice displayed social preference in both stressful (n = 11) and stress reducing (n = 38) conditions (*Figure 2C*). Social preference (*Figure 2D and E*) was measured as time spent in the chamber containing a social object compared to time spent in chamber with a non-social object. These results show that C57BL/6 mice approach a conspecific mouse more than an inanimate object, regardless of the level of experimental stress while BALB/c mice behave this way only when using stress reducing

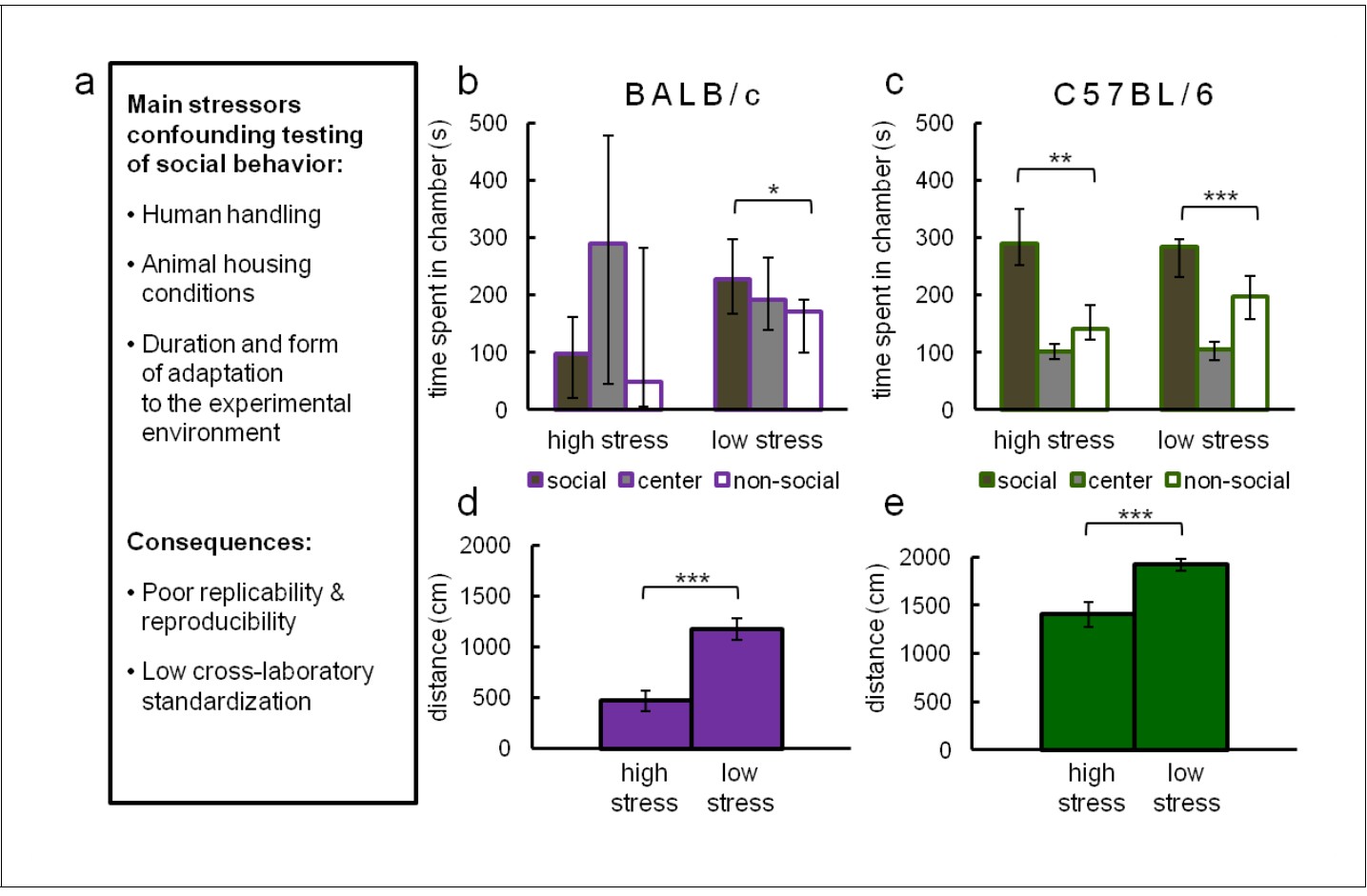

**Figure 2.** Main stressors interfering with reliable measurement of social behavior in rodents (**a**) and their effects on social preference scores in C57BL/6 and BALB/c mice in the conventional three-chambered test (**b–e**) under low- and high-stress conditions. High-stress conditions differ from low-stress conditions in the intensity of light and subjects' and mouse social objects' habituation to the experimenter and the experimental environment (for a detailed protocol see 'Materials and methods'). (**b**) BALB/c mice showed social preference only in low-stress conditions (n = 17), and they avoided social interactions when tested in a typical experimental setting (n = 11). (**c**) In contrast, C57BL/6 mice displayed social preference in both stressful (n = 11) and stress-reducing (n = 38) conditions. Social preference was calculated as the time spent in the chamber containing the social object compared to the time spent in the chamber with a non-social object. (**d,e**) Under stress, both tested strains of mice showed reduced locomotor activity. Data are median values and error bars represent IQR (interquartile range), *p<0.05, **p<0.01, ***p<0.001 (Mann-Whitney U-test).

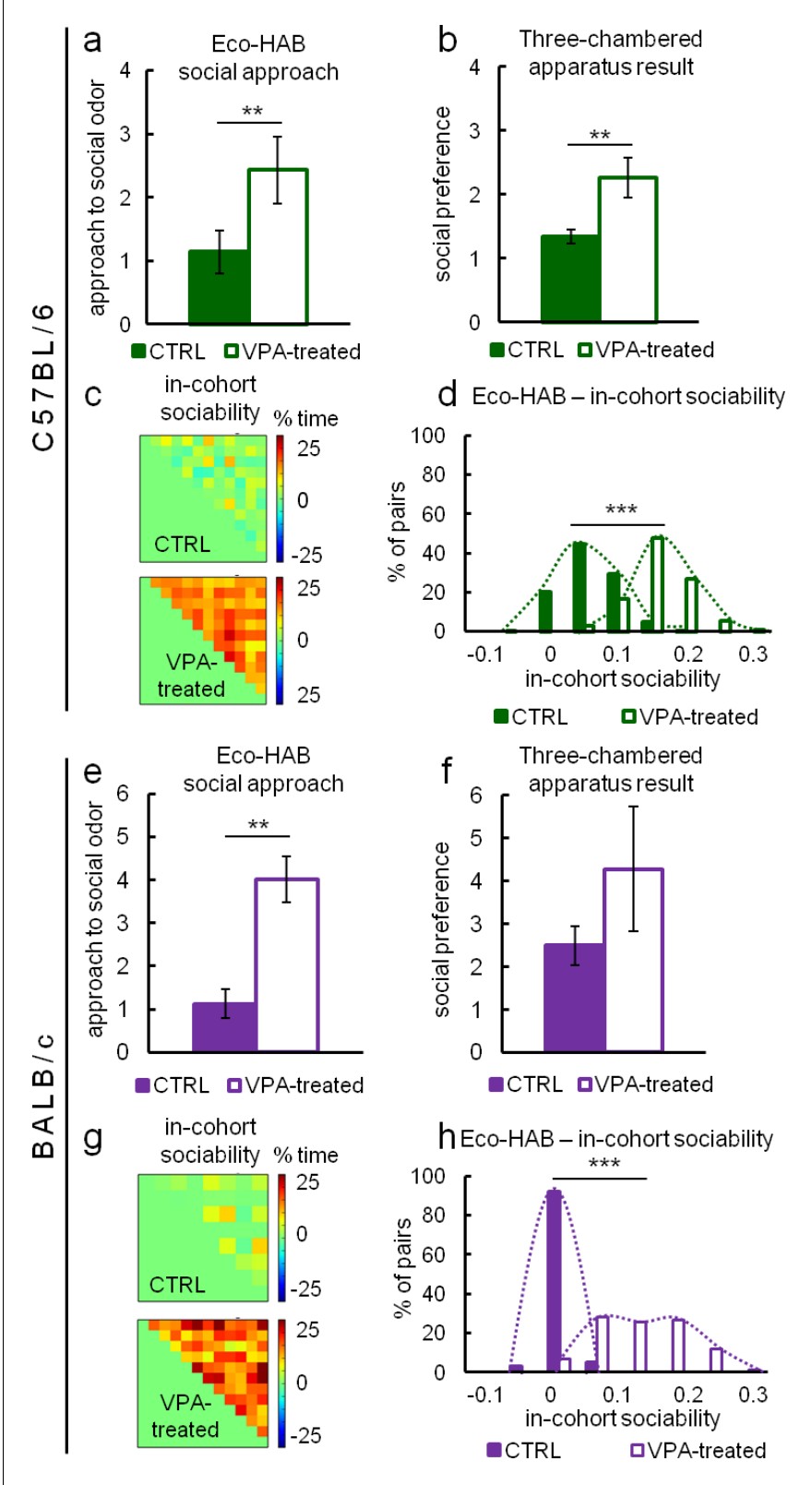

**Figure 3.** Sociability measurements in Eco-HAB and three-chambered apparatus social approach test performed under stress reducing conditions. Figure (**a–d**) depict tests involving C57BL/6 mice and (**e–h**) tests involving BALB/ c mice. (**a**) and (**e**) show social approach in the Eco-HAB system defined as the increase in proportion of time spent in the compartment with social odor during the first hour after its presentation, divided by the proportion of
*Figure 3 continued on next page*

*Figure 3 continued*

time spent in the compartment with non-social stimulus. For (**a**), VPA-treated n = 26 and CTRL n = 35. For (**e**), VPA-treated n = 18 and CTRL n = 20. (**b**) and (**f**) show social approach in the three-chambered test, defined as the increase in proportion of time spent in the compartment with an unfamiliar mouse, divided by the proportion of time spent in the compartment with unfamiliar inanimate object. For (**b**), VPA-treated n = 18, CTRL n = 27. For (**f**), VPA-treated n = 23 and CTRL n = 26. (**c**) and (**g**) show density plot matrices for Eco-HAB housed control and valproate-treated cohorts. Each small square, for which position in the matrix represents one pair of subjects, shows the total time spent together minus the time animals would spend together assuming independent exploration of the apparatus (see 'Materials and methods'). Histograms (**d**) and (**h**) show the distribution of this measure for all pairs of valproate-treated and control animals. Data are mean values and error bars represent SEM, *p<0.05, **p<0.01, ***p<0.001 (Mann-Whitney U-test).

The following source data and figure supplements are available for figure 3:

**Source data 1.** Eco-HAB measured social approach and in-cohort sociability of valproate-treated and control C57BL/6 and BALB/c mice.

**Source data 2.** Eco-HAB measured social approach and in-cohort sociability of valproate-treated and control C57BL/6 and BALB/c mice.

**Source data 3.** Eco-HAB measured social approach and in-cohort sociability of valproate-treated and control C57BL/6 and BALB/c mice.

**Source data 4.** Eco-HAB measured social approach and in-cohort sociability of valproate-treated and control C57BL/6 and BALB/c mice.

**Figure supplement 1.** Three-chambered apparatus testing performed on group-housed valproate-treated C57BL/6 (VPA-treated n = 14, CTRL n = 38) (**a**) and BALB/c (n = VPA-treated n = 15, CTRL n = 17) (**b**) mice did not reveal any differences in sociability.

**Figure supplement 2.** Eco-HAB allows for long-term monitoring of responses to social stimuli.

procedures. In both tested strains, conventional experimental treatment (high stress) reduced loco-motor activity which may have attenuated the number of social contacts and influenced their propensity for exploration.

## Eco-HAB – validation of the method

In order to explore how Eco-HAB data relate to the most commonly used social approach task, we compared our results with the 3ChA test performed under stress-reducing conditions on both group and single-housed subjects. For these tests, social approach in the Eco-HAB system is calculated as the increase in the proportion of time spent in the compartment with social odor during the first hour after its presentation, divided by the proportion of time spent in the compartment with non-social stimulus. In the three-chambered test, social approach is the increase in the proportion of time spent in the compartment with an unfamiliar mouse, divided by the proportion of time spent in the compartment with an unfamiliar inanimate object. We used animals displaying different levels of social interactions, namely valproate-treated (VPA) mice of C57BL/6 and BALB/c strains. Single pre-natal valproate exposure is considered a mouse model of an environmental insult (a potential trigger) contributing to development of autism spectrum disorders (*Roullet et al., 2013*), albeit some recent results report increased sociability of VPA-treated animals (*Štefánik et al., 2015*).

Our results are clearly in favor of valproate increasing sociability in both tested strains. Automated Eco-HAB testing (*Figure 3A*) showed that valproate-treated C57BL/6 mice display increased social approach. Interestingly, 3ChA testing revealed the same result when subjects were single-housed (*Figure 3B*), but no differences between VPA and control animals when subjects were group-housed (*Figure 3—figure supplement 1A*). In BALB/c mice, despite significant attempts at reducing experimental stress in 3ChA testing, only Eco-HAB revealed a significant increase in social behavior caused by VPA (*Figure 3E*). Manual assessment with the use of the 3ChA showed the same

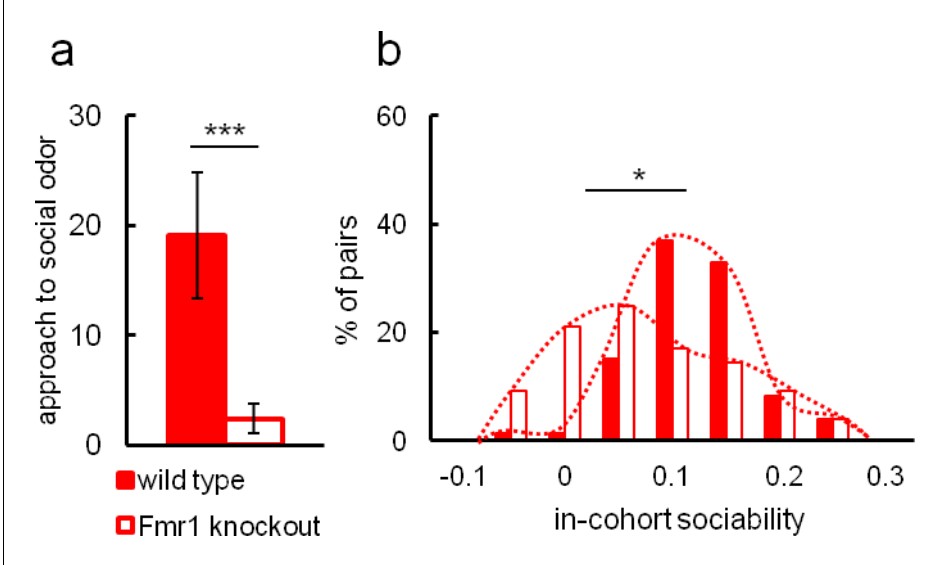

**Figure 4.** Social impairment of *Fmr1* knockout mice compared to wild-type control measured in Eco-HAB. *Fmr1* knockouts, n = 22. Wild-type controls, n = 10. (a) Odor-based social preference in the Eco-HAB system defined as the increase in proportion of time spent in the compartment with social odor during the first hour after its presentation, divided by the proportion of time spent in the compartment with non-social stimulus. Histogram (b) shows the distribution of in-cohort sociability for all pairs of knockout and control animals. Data are mean values and error bars represent SEM, *p<0.05, **p<0.01, ***p<0.001 (Mann-Whitney U-test).

The following source data is available for figure 4:

**Source data 1.** Eco-HAB measured social approach and in-cohort sociability of *Fmr1* knockouts and wild-type controls.

**Source data 2.** Eco-HAB measured social approach and in-cohort sociability of *Fmr1* knockouts and wild-type controls.

trend in single-housed animals; however, differences were blurred by a huge variability in the scores (*Figure 3F*). Again, no differences were found between VPA and control, group-housed BALB/c subjects (*Figure 3—figure supplement 1B*).

Even though Eco-HAB data was consistent with the results of manual tests of sociability performed on single-housed animals (see *Figure 3A,B and E,F*), one must keep in mind that approach behavior or proximity may reflect not only affiliative, but also novelty-seeking, aggressive, or sexual motivation. Thus, a major remaining challenge is to precisely identify the motivation involved in a particular social interaction. This is extremely difficult in one-trial manual experiments, but possible to do in Eco-HAB because of the long monitoring time.

To show that our novel in-cohort measure of sociability agrees with approach to social odor results, we utilized Eco-HAB's capacity to investigate subjects' preferences for spending time together within each cohort (see 'Materials and methods') in VPA animals. Results show that VPA C57BL/6 mice stay together more often than respective controls (*Figure 3C,D*). The same tendency was found for VPA BALB/c animals (*Figure 3G,H*).

In view of the conclusions regarding valproate effects, we further used Eco-HAB to test approach to social odor and in-cohort sociability in *Fmr1* knockout mice (*Figure 4A,B*). These mice are a well-established animal model of autism and have repeatedly been reported to display social deficits (*Bernardet and Crusio, 2006*; *Mines et al., 2010*; *Mineur et al., 2006*; *Santos et al., 2014*; *Sidhu et al., 2014*). *Figure 4A* depicts social approach and 4B a histogram of in-cohort sociability as defined previously for Eco-HAB for both *Fmr1* knockouts (n = 22) and wild-type controls (n = 18). The Fmr1 knockouts display a lower level of social approach and decreased in-cohort sociability as

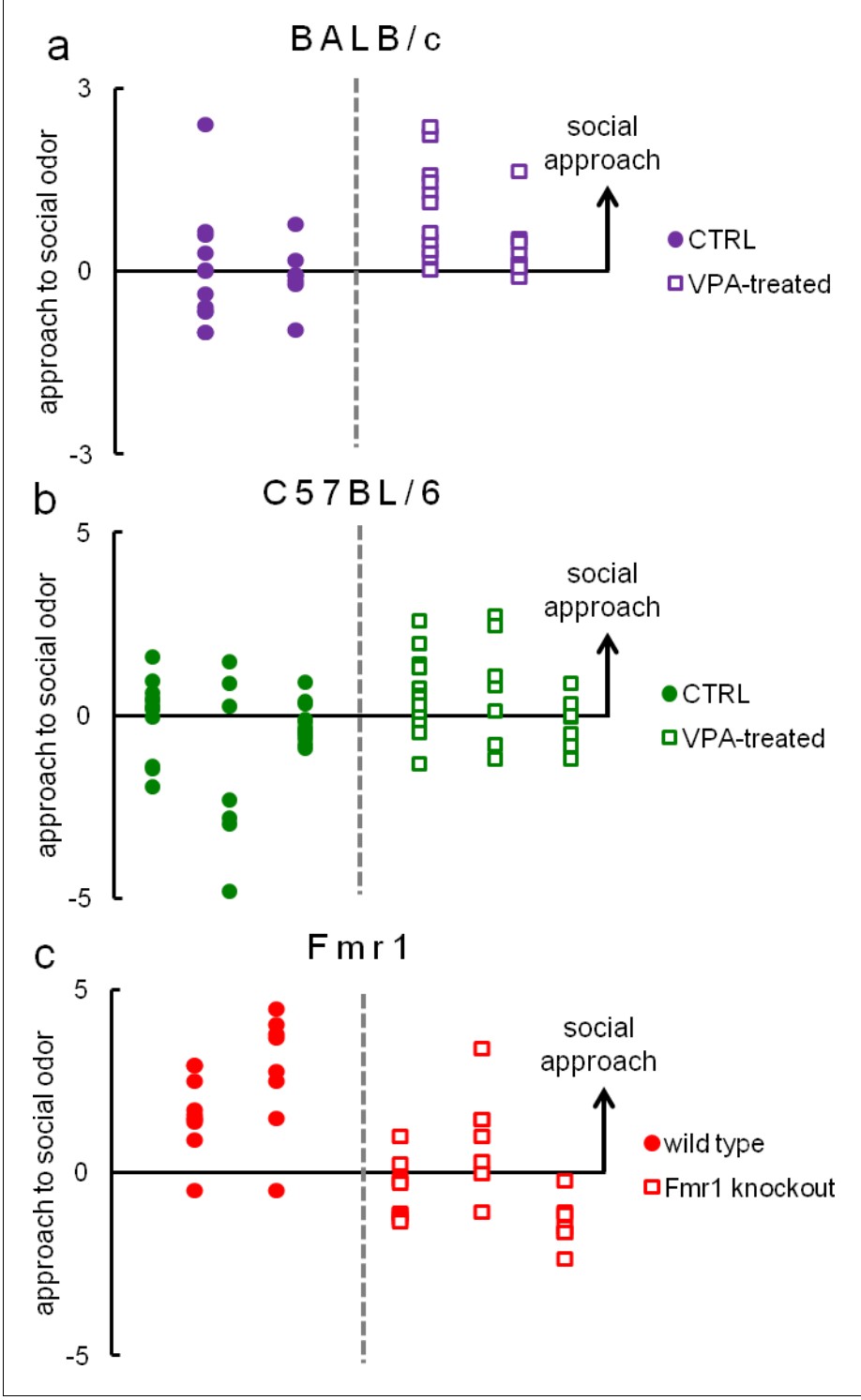

**Figure 5.** Eco-HAB provides reproducible assessment of approach to social odor in group-housed mice. Individual results of approach to social odor for all cohorts of (**a**) valproate-treated (n = 20) and control (n = 18) BALB/c subjects (4 cohorts), (**b**) valproate-treated (n = 26) and control (n = 35) C57BL/6 mice (6 cohorts) and (**c**) Fmr1 knockouts (n = 22) and wild-type (n = 18) animals (5 cohorts). Each column represents one cohort of animals, while data points (dots and squares) represent scores of particular mice. Since the measure of approach to social odor is a proportion (for detailed description see 'Materials and methods'), which may take values from 0 to +∞, we present logarithmic data to depict reproducibility of social preference and social avoidance in an unbiased

*Figure 5 continued on next page*

*Figure 5 continued*

manner. All analyses, including statistical testing, were performed on raw data. Average results of these data are presented in *Figures 3A,E* and *4A*, respectively.

The following source data is available for figure 5:

**Source data 1.** Eco-HAB measured social approach score for valproate-treated and control C57BL/6 and BALB/c mice and Fmr1 knockouts and wild-type controls.

**Source data 2.** Eco-HAB measured social approach score for valproate-treated and control C57BL/6 and BALB/c mice and Fmr1 knockouts and wild-type controls.

**Source data 3** Eco-HAB measured social approach score for valproate-treated and control C57BL/6 and BALB/c mice and Fmr1 knockouts and wild-type controls.

compared to wild-type. The results clearly confirm the impairment of social behavior in *Fmr1* knockouts, as has been observed previously.

## Reproducibility of Eco-HAB data

Since replication failure is one of the main issues in conventional tests of social behavior, to illustrate reproducibility of Eco-HAB's measures, we compared individual scores of social odor approach for all mice within 10 cohorts of VPA-treated and control BALB/c mice (*Figure 5A*, 20 VPA-treated and

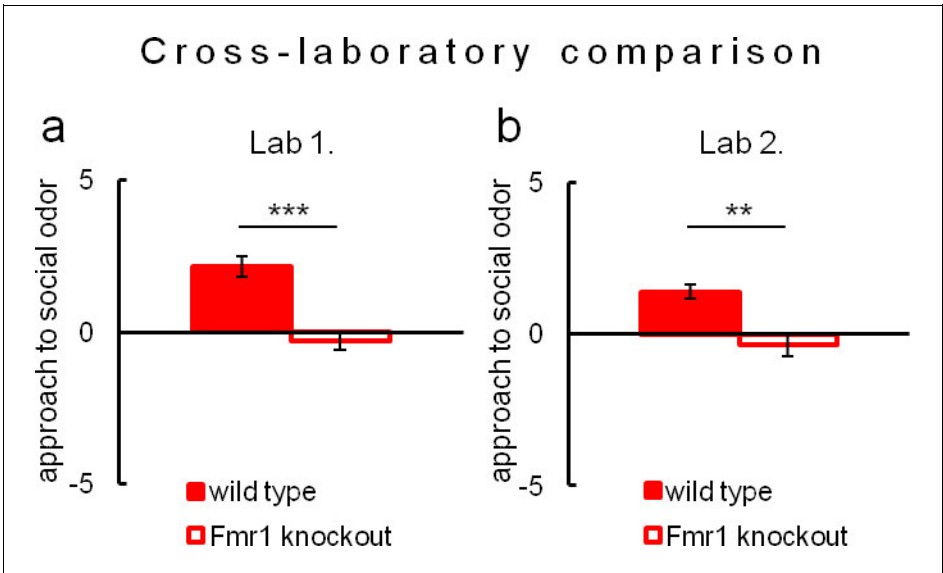

**Figure 6.** Assessment of approach to social odor in Fmr1 knockouts and respective littermate controls performed in two different laboratories. (a, Fmr1 knockout n = 22, wild-type control n = 18) vs. (b, Fmr1 knockout n = 11, wild-type control n = 9). Regardless of experimental environment, evaluation carried out in Eco-HAB revealed comparable impairment in Fmr1 knockouts. Presented data are logarithmic values.

The following source data is available for figure 6:

**Source data 1.** We include source data for *Figures 6*, *7* and *8* concerning reproducibility results of both Eco-HAB measures.

**Source data 2.** We include source data for *Figures 6*, *7* and *8* concerning reproducibility results of both Eco-HAB measures.

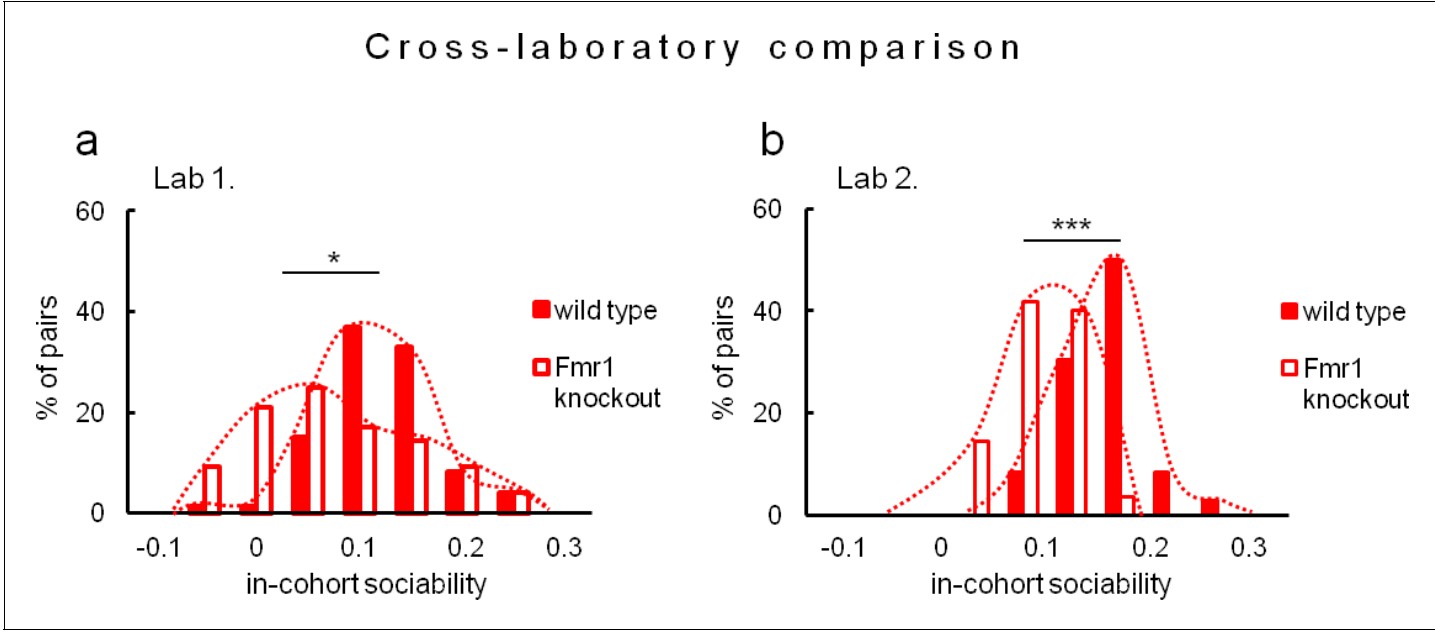

**Figure 7.** Evaluation of in-cohort sociability in Fmr1 knockouts and wild-type littermate controls undertaken in two different laboratories. (a, Fmr1 knockout n = 22, wild-type control n = 18) vs. (b, Fmr1 knockout n = 11, wild-type control n = 9) – gives corresponding results. A histogram illustrating score of Fmr1 knockouts is shifted to the left as compared to that for wild-type control, signifying less time voluntarily spent together with other subjects within a tested cohort.

The following source data is available for figure 7:

**Source data 1.** We include source data for *Figures 6*, *7* and *8* concerning reproducibility results of both Eco-HAB measures.

**Source data 2.** We include source data for *Figures 6*, *7* and *8* concerning reproducibility results of both Eco-HAB measures.

18 controls), another 10 cohorts of C57BL/6 mice (*Figure 5B*, 26 VPA-treated and 35 controls), as well as 4 cohorts of Fmr1 knockout (n = 22) and wild-type animals (n = 18, *Figure 5C*).

Due to evident deficits of social behavior in Fmr1 knockouts, we chose this model to further investigate predictability of phenotyping performed under different environmental conditions. To that end we repeated evaluation of sociability in Fmr1 knockout animals and respective controls in another laboratory. Results confirm social impairments of Fmr1 knockouts and show that both standardized Eco-HAB measures, approach to social odor (*Figure 6*) and in-cohort sociability (*Figure 7*) were highly reproducible in those two independent studies. Further, to test if the individual sociability measure – approach to social odor – is stable in particular subjects, we performed two subsequent replications of Eco-HAB testing in the same cohorts of Fmr1 knockout and wild-type mice. Experiments were separated by a 10-day period of regular housing. Within-subject comparison (*Figure 8*) reveals high reproducibility of sociability assessment in particular subjects over time. Taken together, these studies show that Eco-HAB is a reliable tool that is reproducible for a number of tasks, from assessment of individual sociability, through phenotyping of subsequent cohorts of mice, to cross-laboratory comparisons.

## Eco-HAB measurement is unbiased by social hierarchy and allows for long-term monitoring of social behavior

Social hierarchy, occurring in group-housed mice, could interfere with social behavior measures. For example, dominant mice may occupy territories and restrict the exploration of others. To test this hypothesis, following Eco-HAB testing, we performed a U-tube dominance test (*Lindzey et al., 1961*). In this test, mice were repeatedly placed facing another mouse from a tested cohort in a narrow tube. We show that winner/loser scores were not associated with activity-based exploration of the available territory (*Figure 9*).

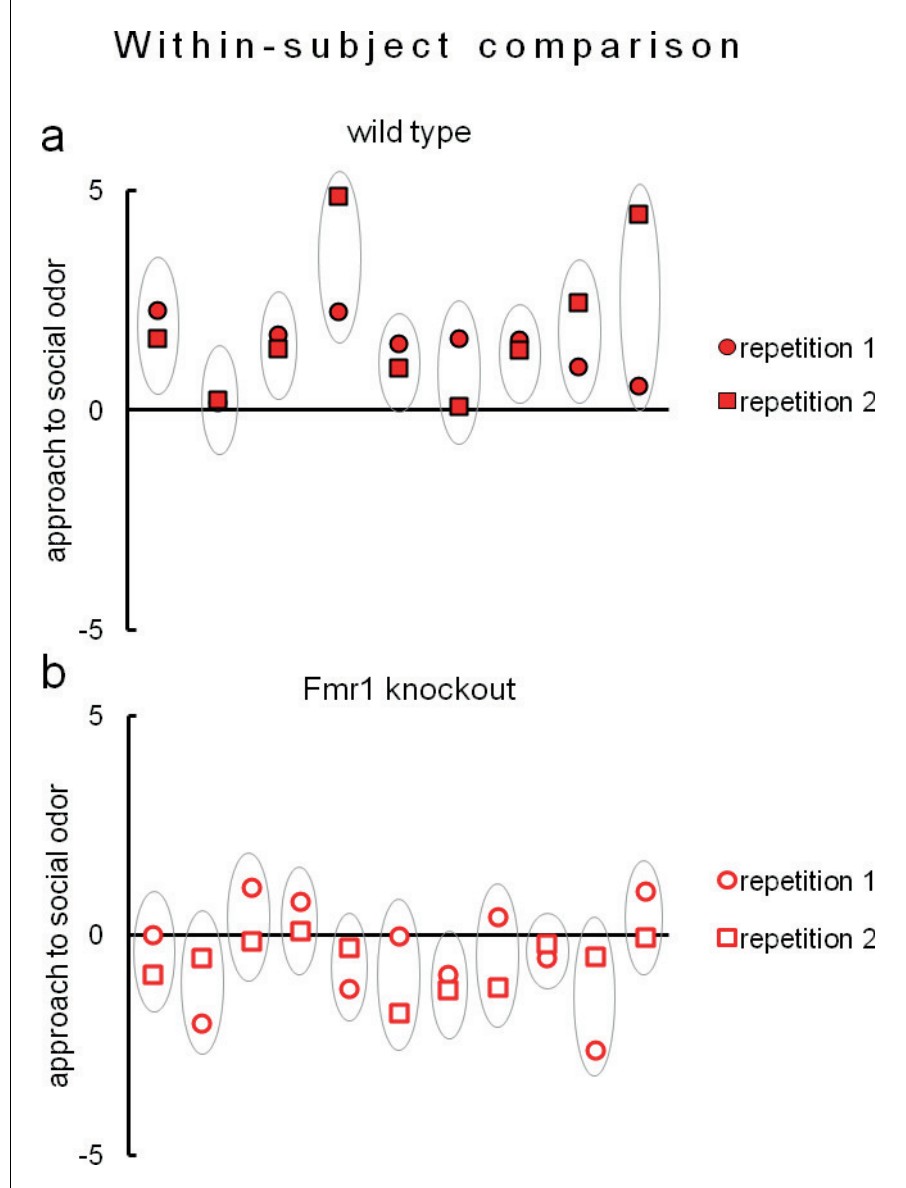

**Figure 8.** Eco-HAB allows remarkably reproducible assessment of approach to social odor in both (**a**) wild-type mice (n = 9) and (**b**) Fmr1 knockouts (n = 11). Evaluation of social behavior of subjects was repeated twice in identical Eco-HAB experiments, separated by a 10-day period of regular housing. Each aligned dot and square encircled by an oval represent individual score of approach to social odor for each tested mouse, measured in two subsequent experimental repetitions. Dots are data, while the ovals serve to guide the eye. Data presented are logarithmic values.

The following source data is available for figure 8:

**Source data 1.** We include source data for *Figures 6*, *7* and *8* concerning reproducibility results of both Eco-HAB measures.

**Source data 2.** We include source data for *Figures 6*, *7* and *8* concerning reproducibility results of both Eco-HAB measures.

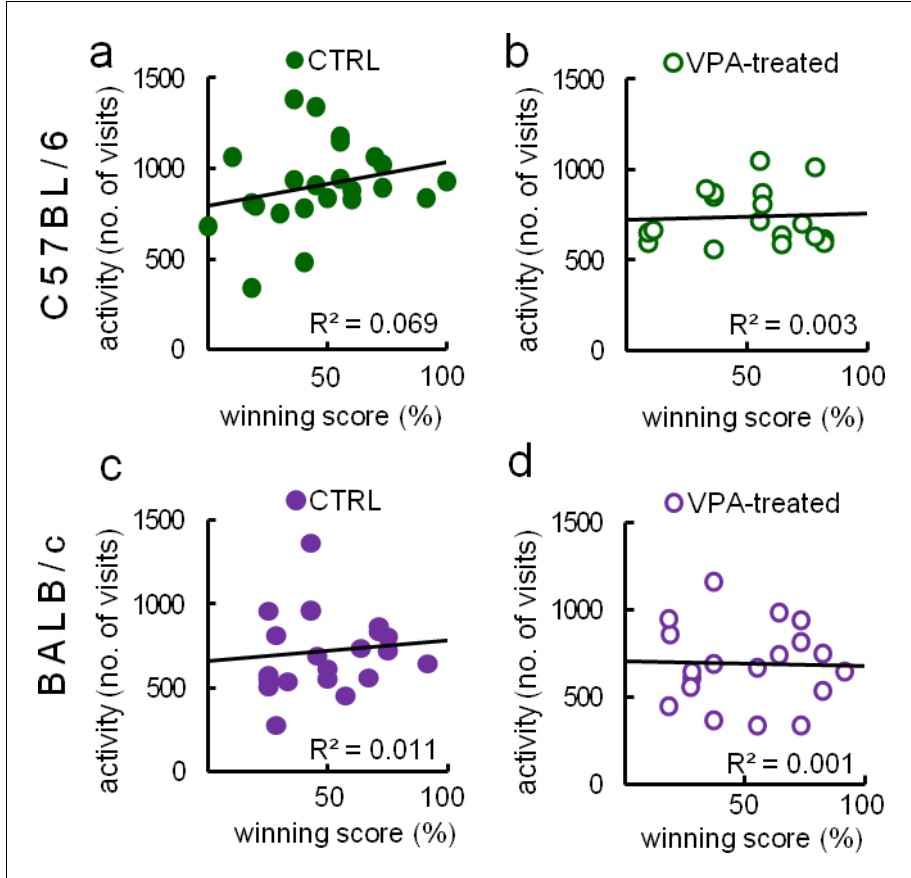

**Figure 9.** Tube dominance score of an animal does not correlate with overall activity in Eco-HAB apparatus. Dominance is expressed as percentage of won encounters ('winning score percent') in the U-tube dominance test (see 'Materials and methods'). Activity in Eco-HAB is defined as the number of visits to all of its compartments during the first 12 hr of the habituation period. Social hierarchy does not correlate with exploration of the territory in either of the tested groups: (a) control (n = 23) or (b) valproate-treated C57BL/6 mice (n = 26), (c) control (n = 32) or (d) valproate-treated BALB/c mice (n = 19). Dependence between two variables tested by Pearson product-moment correlation coefficient.

The following source data and figure supplement are available for figure 9:

**Source data 1.** Raw data from U-tube dominance test and Eco-HAB measured activity (number of visits to all compartments of the apparatus during 1st 12-hr period of adaptation).

**Source data 2.** Raw data from U-tube dominance test and Eco-HAB measured activity (number of visits to all compartments of the apparatus during 1st 12 hr period of adaptation).

**Figure supplement 1.** Aggressive interactions during testing in Eco-HAB are rare regardless of the tested strain.

## Conclusion

Eco-HAB is an open-source system which combines novel elements to provide low-stress experimental settings for high-throughput, automatic testing of conspecific-related behavior in mice. The testing environment is spacious, resembles the natural habitat of mice, and exploits innate behavioral patterns of this species to test relevant aspects of mouse sociability. Unlike short-term assessment using manual tests, Eco-HAB allows for long-term monitoring of social behaviors. Importantly, data collected in Eco-HAB show that the dynamics of response to social stimuli may differ depending on the tested strain of mice (see *Figure 3—figure supplement 2*).

Individual tracking with RFID technology allows testing of group-housed animals without human handling – the most important confounding factor in manually conducted tests. In contrast, manual methods have so far allowed for testing isolated animals in relatively small cages in an environment unfamiliar to tested subjects. Existing automated systems, even though technologically advanced and applicable for collection of large datasets, either lack ecological pertinence or are not suitable for assessment of relevant social behaviors in mice. All these problems are addressed by Eco-HAB.

In contrast to available open-arena set-ups, the apparatus we constructed allows mice to display their natural affiliation patterns. As shown by *Weissbrod et al. (2013)*, open-arena testing entails frequent display of aggressive behaviors such as chasing or fighting. Based on knowledge from this and similar studies (*de Chaumont et al., 2012*; *Shemesh et al., 2013*), we concentrated on creating an environment that would reduce these types of interactions in order to study different types of social behavior, namely stable affiliations among mice of the same sex. The separate and dispersed sub-territories of the Eco-HAB, resembling borrows inhabited by mice in the wild, alleviate extensive territorial fighting (see *Figure 9—figure supplement 1*) and prompts animals to form sub-groups in accordance with their natural preferences.

Experiments with larger groups of animals in an undisturbed setting provide access to information unavailable when isolated animals are tested. In addition to providing a platform for measuring in-cohort sociability and scent based social approach, the Eco-HAB and its associated software has proven very useful for in-depth analysis of individual social behaviors. Heat maps of mouse pair inter-actions may be used to identify more and less social individuals – data useful for comparison with other measures (i.a. individual differences in specific genes or neural markers). Heat maps may also reveal particular littermate affinities or be used to assess stable affiliations between mates or same-sex peers. One can also envision performing experiments on sub-groups of treated/untreated co-housed mice, where differences could be assessed within a single testing session. Compared to testing experimental and control groups separately, experiments on relevant populations allow evaluation of whether the social environment is an essential factor influencing littermate-related behavior. One can also envision expanding these measurements beyond mice to include other rodents, such as prairie or meadow voles.

A noteworthy asset of the Eco-HAB apparatus is the free, custom software which aids in obtaining effective measurements and speeds up data analysis. Appropriate programs were created for the purpose of data collection and conversion as well as in-depth evaluation of social behaviors. This code is open source and can be expanded to encompass new analyses. Assessing reliability of differ-ent behavioral measures, we chose the most valid. Nevertheless, in the present form, our system does not allow for the recognition of particular types of subtle littermate-related behaviors that might skew results such as having two animals in the same chamber but facing away from each other and not interacting. While it is not possible to distinguish different types of social interactions yet, casual observations of video recordings obtained during numerous experiments lead us to believe that such events are rather accidental.

In summary, compared to manual tests of sociability, our system provides more reliable data, faster, and with less manpower for several key behavioral measures.

## Materials and methods

### Animals

Animals were treated in accordance with the ethical standards of the European Union (directive no. 2010/63/UE) and Polish regulations. All experimental procedures were pre-approved by the Local Ethics Committee. Valproate-treated mice of C57BL/6 and BALB/c strains as well as *Fmr1* knockout mice of the FVB strain (RRID:IMSR_JAX:008909) and all respective littermate controls were bred in the Animal House of the Faculty of Biology, University of Warsaw.

The effects of prenatal exposure to valproic acid (VPA) were assessed for C57BL/6 and BALB/c strains of animal. To do this, mice were mated with other mice of the same strain and pregnancy was confirmed by the presence of a vaginal plug on embryonic day 0 (E0). On E13, pregnant females received a single subcutaneous injection of 600 mg/kg VPA (Sigma-Aldrich) dissolved in saline. The concentration of the drug in saline was 58–63 mg/ml. The volume of the injected fluid was < 0.35 ml

to facilitate proper absorption of the solution. Behavioral experiments were performed in male 2.5- to 5-month-old offspring. Animals' age was balanced across experimental conditions.

Depending on the experiment, animals were group or single housed with a 12 hr/12 hr light/dark cycle with water and food provided ad libitum. In housing and experimental rooms, the temperature was maintained at 23–24°C with humidity levels between 35% and 45%. In order to reduce aggression in BALB/c group-housed males, we enriched the pre-experimental environment and utilized rat-sized cages to help decrease territorial behaviors. Overtly aggressive BALB/c males were removed from the group cages and were not used in further procedures. As male mice of FVB strain are extremely territorial, it was difficult to eliminate aggressive behaviors that occurred in group-housing. For that reason, only female *Fmr1* knockouts and littermate controls (2.5- to 4-month-old) were utilized in behavioral experiments. Animals' age was balanced across experimental conditions.

The multiplicities of the animal cohorts were chosen following our previous work (*Puścian et al., 2014*), in which we determined optimal parameters for the measurement of spontaneous reward-motivated behavior in socially enriched environments and we discussed the number of biological replications required to establish whether a given behavioral parameter is sufficiently reproducible. In the present study, we performed all the analyses in accordance with our previous findings and taking into account the area of Eco-HAB system.

## Three-chambered apparatus testing

This assay consisted of an experimental box (length – 620 mm, width – 425 mm, height – 250 mm) divided into three equally sized areas. The middle area was object free, while the side areas contained either a social or a non-social stimulus placed in small steel cages (length - 95 mm, width - 95 mm, height - 105 mm). The protocol for assessment of social preference consisted of three sessions: exploration of the middle chamber, exploration of the side-chambers with empty steel cages, and a testing session when social and non-social stimuli were presented. Each session lasted 10 min and was video-recorded. For these experiments, the social stimulus was an unfamiliar mouse of the same strain, sex, and age while the non-social stimulus was a novel blue plastic laboratory bottle cap.

For the purpose of obtaining reliable, undistorted measurements of social preference, a number of steps were taken to minimize stress in the tested animals. Mice were habituated to the experimenter and handling procedures for 14 days prior to testing. All animals used as social stimuli were also subjected to the sham experimental procedure (sitting inside of the steel cage placed in one of the side chambers for 10 min) for 7 days prior to testing day. Transportation of the subjects always took place at least 1 hr before the beginning of the experiment. Procedures were observed and remotely recorded by the experimenter from a separate room to minimize human interference with behavior. Finally, all tests were performed in dimmed light conditions (4–5 lux as measured at the bottom of the apparatus).

We excluded data from the analysis corresponding to the situation in which an animal had not visited both side chambers in either adaptation or social preference testing phase. In other words, we discarded those rare cases in which the animals' locomotor activity was extremely low. In the course of a 3-year period, we performed at least two biological replications of the three-chambered apparatus testing of each experimental and control group.

## Tube dominance test

A dominance test (*Lindzey et al., 1961*) was performed in a U-shaped tube (length – 800 mm, diameter – 33 mm) whose diameter was sufficiently small to prevent animals from turning while inside of it. Two subjects were simultaneously released into the tube at opposite ends and allowed to compete head-to-head until the more dominant subject completely pushed its opponent out of the tube. Forcing the other mouse out of the tube with all four paws was defined as winning. All animals subjected to the procedure were tested in a round-robin system (on average 10 parings per subject) against fellow members of their previously Eco-HAB tested cohort. A dominance score for each individual was calculated as a percentage of confrontations won.

## Eco-HAB – apparatus construction

All elements were either made of Plexiglass (tube-shaped corridors, perforated partitions) or polycarbonate (housing compartments). Housing compartments (length – 250 mm, width – 250 mm, height – 150 mm) had round bottom edges (as in standard housing cages) and two neighboring sidewalls of each compartment were equipped with holes (diameter – 420 mm and 50 mm from the bottom of the compartment) for corridors. Tube-shaped corridors had the following dimensions: length – 300 mm, inner diameter – 40 mm, outer diameter 420 mm. Perforated partitions were rectangular in shape (width – 115 mm, height – 150 mm) with round bottom edges. Perforations were vertical, rectangular shaped, had soft edges, and began 10 mm above the bottom of the cage. Perforations had the following dimensions: height – 130 mm, width 5 mm, with equal 10 mm spacing between perforations. Circular antennas were placed around corridors 40–60 mm from the side walls of housing compartments. Square, stainless steel lids (250 mm long and wide) were fitted for tops of the housing compartments. The two lids placed above the compartments containing perforated partitions were flat, whereas the other two had grid trays for food and water bottles. Illumination levels were standardized between all compartments in an effort to reduce light impact on mouse activity. In the compartments with perforated partitions, light intensity was maintained between 79 and 84 lux. In the compartments equipped with food and water, light intensity was between 18 and 25 lux in shadowed areas and between 75 and 80 lux in non-shadowed areas. It is worth mentioning that the Eco-HAB design is easily scalable. With the exception of RFID coils, whose size needs to be adjusted in accordance to the corridor diagonal, there is no need to change applied electronics or software should one want to adapt the system for species other than mice.

## Eco-HAB testing

To individually identify animals in Eco-HAB, all mice were subcutaneously injected with glass-covered microtransponders (9.5 mm length, 2.2 mm diameter, RF*IP* Ltd) under brief isoflurane anesthesia. Microtransponders emit a unique animal identification code when in range of RFID antennas. After injection of transponders, subjects were moved from the housing facilities to the experimental rooms and adapted to the shifted light/dark cycle of their new environment (the dark phase shifted from 20:00 – 8:00 to 13:00 – 01:00 or 12:00 – 24:00 depending on summer/winter UTC+01:00). For 2 to 3 weeks prior to the behavioral testing, subjects were housed together and grouped appropriately for their respective experiment.

Cohorts consisting of 7 to 12 mice were subjected to 72-hr Eco-HAB testing protocols divided into an adaptation phase (48 hr) and odor-based social preference (approach to social odor) testing phase (24 hr) with access to food and water unrestricted throughout. During the adaptation phase, mice could freely explore all compartments. During the social preference testing phase, olfactory stimuli including either bedding from the cage of an unfamiliar mouse of the same strain, sex and age (novel social scent) or fresh bedding from a different room (novel non-social scent), were presented. Olfactory stimuli were simultaneously placed behind the perforated partitions of opposite testing compartments (see *Figure 1*). Mice could freely explore both testing compartments for 24 hr. Values from the first hour after presentation of the stimulus were used for statistical analysis. We excluded data from the analysis corresponding to the situation in which an animal had not visited chambers where olfactory stimuli had been presented in either adaptation or odor-based social preference testing phase, i.e. we discarded those rare cases in which the animals' locomotor activity was extremely low. In the course of a 2-year period, we performed at least two biological replications of Eco-HAB testing of each experimental and control group. Although the protocol utilized here lasted 72 hr, it is noteworthy that, when required, Eco-HAB can run continuously even for months with only short technical breaks for cage cleaning (every 7 days). The cages are not connected to the IVC system, so air exchange is not an issue. Moreover, any kind of olfactory stimuli may be presented behind perforated partitions, enabling experimenters to investigate matters such as motivational conflicts (e.g. when animals are allowed to choose between sniffing an odor related to food or the scent of a female).

## Eco-HAB – applied electronic solutions

The RFID system for accurately logging mouse position (block schematic diagram presented in *Figure 1—figure supplement 5*) operates as follows: once an RFID chip (microtransponder) arrives in

the proximity of a particular antenna (RFID coil in *Figure 1—figure supplement 5*), the corresponding RFID receiver decodes the message from the chip and then passes it to a microcontroller. The microcontroller extracts the ID number from the decoded message and transmits it via USB to a computer. The computer receives data packets, including ID numbers, from each antenna via virtual serial ports, and writes them to file using dedicated software described in the next section.

The RFID system for Eco-HAB consists of eight RFID receivers working at a frequency of 125 kHz. The system is based on COTS modules plugged into bread boards. The RFID reader module (UN-MOD3) uses an EM4095 receiver coupled to an ATTINY AVR microcontroller. The microcontroller controls the receiver via SPI interface and extracts the ID number of the RFID transponder. This number is then transmitted to the serial-to-USB transceivers via UART. The computer recognizes the transceivers as virtual ports. Since, by default, all RFID receivers work asynchronously with local ceramic oscillators, they interfere with each other and end up disabling operation of the system. To avoid this, the output of a 4 MHz crystal oscillator powered by 5V is distributed to all EM4095 IC clock inputs. All USB to serial modules are attached to an USB hub, which also supplies their power.

To assess the efficiency of the implemented RFID antennas, we compared the Eco-HAB recognition system with manual video-based scoring. We have assessed the number of antenna crossings during the first 6 hr period of the dark phase at the beginning of the adaptation phase, when animals are most active and intensely explore a new environment. Videos were recorded in complete darkness. A source of infra-red light placed above the apparatus was used to illuminate the field of view. Measurements were repeated twice on two independently tested cohorts of mice for the purpose of more reliable assessment. Based on these measurements, we concluded that the implemented system of coils can recognize subjects with high sensitivity, far exceeding that obtained with standard video-recognition methods. Due to the built-in internal synchronization system, RFID antennas run independently and do not disrupt each other. All eight coils may be activated simultaneously for an unlimited time.

## Eco-HAB.rfid – software package for data collection

Eco-HAB.rfid software is able to simultaneously receive and process transponder codes from up to eight RFID antennas. The software defines the end of the read-out as a period of more than 210 ms without a transponder code transmitted to the computer. Eco-HAB.rfid records all read-outs in a plain text file with tab separated columns. Each line consists of an event number, date, time (ms), number of the activated RFID antenna, duration of the transponder read-out (ms), unique transponder code (14 digits), and tag name (if specified in a separate text file – rfid_tags.txt). For the convenience of further data analysis, a single output file contains data from exactly 1 hr of the system's operation. All files are automatically named and saved to an assigned hard drive. Eco-HAB.rfid is written in Delphi programming language and compiled with Borland Delphi 7.0 (Embarcadero Technologies) using ComPort Library ver. 4.11.

## Eco-HAB.py – software package for data processing and analysis

Along with the behavioral assessment system, we designed and created a software package for analyzing data collected by Eco-HAB.rfid software written in the Python programming language (Python 2.7 with NumPy and SciPy libraries). It consists of functions for loading and merging raw data previously stored on the hard drive as well as for converting this data into a series of visits (referred to here as sessions) of each mouse to each of the four Eco-HAB compartments.

### Data processing algorithm

For a mouse to be considered to have spent time in a given compartment, it must first enter and then exit that compartment. The time interval between entrance and exit is its residence time in the area. Occasionally, mice pass an antenna too swiftly to trigger a recordable event or pass swiftly through a compartment without lingering. The following filtering procedure was used to automatically account for these types of occasions:

1. For each pair of events the following conditions are checked:
2. Events are considered pairwise in consecutive order, e.g., first (e1, e2), then (e2, e3), and so on until (eN-1, eN).
3. For each mouse we make a list of all events e1, e2,... eN.

a. If the time between the two events is less than a given threshold (we used 2 s), the pair is skipped. Such signals are either multiple readings of a transponder by the same antenna (an animal lingering under an antenna) or events triggered by an animal running through a compartment rather than staying in it.
b. If both events were recorded at the same antenna, and the time between them is greater than the threshold, we determine that the mouse spent that time in the nearest compartment (the animal entered a compartment and left through the same corridor).
c. If both events were recorded in the same corridor, the pair is skipped. This indicates an animal moving through a corridor.
d. If the two events were recorded in two different corridors adjacent to a compartment, this indicates that the mouse spent the intervening time in the connecting compartment.
e. If the two events were recorded in the opposite (parallel) corridors, the pair is skipped. This is a very rare event (<0.6% of pairs above the 2-s threshold) because it can only happen if a mouse passes at least two antennas unnoticed.

The goal of this algorithm is to produce a list of sessions having associated start times, end times, compartment numbers, RFID transponder numbers, and a flag indicating whether antennas registered were consecutive or not. All ambiguous events (those found in steps 3a or 3e) are filtered out, leaving a final list of reliable events for further analysis.

## Using the scripts

To facilitate data loading and processing, Python scripts EcoHab.py and ExperimentConfigFile.py are provided. Three classes are defined in these scripts: EcoHabData, EcoHabSessions, and ExperimentConfigFile.

Raw text data files are loaded and merged into an EcoHabData object:

ehd = EcoHabData(path_to_data)

Once the object is created, various attributes of the raw data can be accessed. For instance, if mice is a list of one or more transponder numbers, then the list of antennas which detected them can be retrieved using ehd.getantennas(mice) and the times at which they were detected (event times) can be retrieved using ehd.gettimes(mice).

An EcoHabSessions object, containing the sessions of animals in Eco-HAB compartments, is created from an EcoHabData object:

ehs = EcoHabSessions(ehd)

This generates the list of sessions using the data processing algorithm described above. In this new, filtered data object, specific attributes of the sessions can be retrieved. The functions ehs.getaddresses(mice), ehs.getstarttimes(mice), ehs.getendtimes(mice), and ehs.getdurations(mice) return compartment numbers, start times, end times, and durations of sessions, respectively. Data from specific time intervals can be selected by the masking function ehs.mask_data. For example, calling ehs.mask_data(t1, t2) retrieves only those sessions starting after t1 and before t2. It is often convenient for an experimenter to mask data according to specific, well-defined experimental phases. Rather than having to calculate and remember numerical time values at which a given phase started and stopped for numerous phases across many experiments, these values can be quickly defined in a text file. This file can be then used to mask multiple data sets according to easily remembered phase names. Such a file is named config.txt and contains start and stop points for each phase an experimenter wishes to define in the following format:

[ADAPTATION - 1. dark phase]
startdate = 16.02.2015
starttime = 12:00
enddate = 17.02.2015
endtime = 00:00

Once such a configuration file has been made, it can be read using the ExperimentConfigFile class:

cf = ExperimentConfigFile(path_to_data)

The configuration object cf can now be used for masking data:

ehs.mask_data(*cf.gettime('ADAPTATION - 1. dark phase'))

This will limit future responses to sessions starting during the 'ADAPTATION - 1. dark phase' and provides a powerful tool for automated processing of multiple data sets. Both scripts for data analysis together with an example data set are provided in Materials and methods.

## Approach to social odor

To calculate the approach to social odor of a given animal, total time spent in the compartment containing a social olfactory stimulus, $T_S$, and total time spent in the compartment with a non-social olfactory stimulus, $T_{nS}$, are separately calculated for a select time bin following presentation of olfactory cues. Similar values, $t_S$ and $t_{nS}$, are then calculated for a time bin selected from the last dark phase before presentation of stimuli (baseline conditions). The approach to social odor was then calculated as the ratio of $T_S/T_{nS}$ to $t_S/t_{nS}$.

## In-cohort sociability

The in-cohort sociability of each pair of mice within a given cohort is a measure of sociability that is unique to Eco-HAB system. For a particular pair of subjects, animal a and animal b, we first calculate the times spent by the mice in each of the four compartments during a chosen experimental period: $t_{a1}$, $t_{a2}$, $t_{a3}$, $t_{a4}$ for animal a, and $t_{b1}$, $t_{b2}$, $t_{b3}$, $t_{b4}$ for animal b. The total time spent by the pair together in each of the cages is also calculated: $t_{ab} = t_{ab1} + ;t_{ab2} + t_{ab3} + t_{ab4}$. All times are normalized by the total time of the analyzed segment, so that each of the quantities fall between 0 and 1. The in-cohort sociability (*Figure 1—figure supplement 2c*) is then defined by $t_{ab} - (t_{a1}*t_{b1} + t_{a2}*t_{b2} + t_{a3}*t_{b3} + t_{a4}*t_{b4})$, which is the total time spent together (*Figure 1—figure supplement 2a*) minus the time animals would spend together assuming independent exploration of the apparatus (*Figure 1—figure supplement 2b*).

## Measurement of aggressive encounters in Eco-HAB

We assessed the number and cumulative time of aggressive encounters (fighting, chasing and biting) for every strain tested in Eco-HAB by manual video-based scoring. Behaviors were measured during the first 6 hr period of the dark phase at the beginning of the adaptation period. This is the time when animals are most active, intensely exploring their new environment, and when aggressive behaviors were most probable as unstable social relations on novel territory are being tested by the animals.

## Statistical analysis

Statistical analyses of raw data were performed with Statistica 8.0 (StatSoft) and GraphPad Prism6 software. None of the presented datasets met the criteria for parametric analyses and were therefore subjected to non-parametric testing with the Mann-Whitney U-Test. The criterion for statistical significance was a probability level of $p < 0.05$.

## Acknowledgements

We are thankful to Thomas G Custer, Ksenia Meyza and Krzysztof Turzynski for critical reading of the manuscript and to Karolina Rokosz for advising us on figures' design.

## Additional information

### Funding

| Funder | Grant reference number | Author |
| --- | --- | --- |
| Swiss Contribution to the enlarged European Union | PSPB-210 | Alicja Puścian<br>Hans-Peter Lipp<br>Ewelina Knapska |
| Narodowe Centrum Nauki | 2013/08/W/NZ4/00691 | Szymon Łęski<br>Grzegorz Kasprowicz<br>Ewelina Knapska |

The funders had no role in study design, data collection and interpretation, or the decision to submit the work for publication.

## Author contributions

AP, Conception and design, Acquisition of data, Analysis and interpretation of data, Drafting or revising the article, Critically read and approved the final version of the manuscript; SŁ, Conception and design, Analysis and interpretation of data, Critically read and approved the final version of the manuscript; GK, MW, JB, TN, PMB, Acquisition of data, Critically read and approved the final version of the manuscript; H-PL, Drafting or revising the article, Critically read and approved the final version of the manuscript; EK, Conception and design, Analysis and interpretation of data, Drafting or revising the article, Critically read and approved the final version of the manuscript

## Author ORCIDs

Alicja Puścian, http://orcid.org/0000-0002-7029-1275
Szymon Łęski, http://orcid.org/0000-0003-1764-1907
Paweł M Boguszewski, http://orcid.org/0000-0002-7210-6950
Ewelina Knapska, http://orcid.org/0000-0001-9319-2176

## Ethics

Animal experimentation: Animals were treated in accordance with the ethical standards of the European Union (directive no. 2010/63/UE) and Polish regulations. All experimental procedures were pre-approved by the Local Ethics Committee no. 1 for the city of Warsaw. Permits' numbers 207/2011, 361/2012, 371/2012, 560/2014.

# Additional files

## Supplementary files

• Supplementary file 1. Eco-HAB.py scripts with sample data enabling their execution.

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
