## [Decision Letter]

[Editors’ note: a previous version of this study was rejected after peer review, but the authors submitted for reconsideration. The first decision letter after peer review is shown below.]

Thank you for submitting your work entitled "Eco-HAB – fully automated and ecologically relevant assessment of social impairments in mouse models of autism" for consideration by *eLife*. Your article has been favorably evaluated by a Senior Editor and four reviewers, one of whom, Peggy Mason, is a member of our Board of Reviewing Editors. The following individuals involved in the review of your submission have agreed to reveal their identity: Thomas Bourgeron (peer reviewer). Our decision has been reached after consultation between the reviewers. Based on these discussions and the individual reviews below, we regret to inform you that your work will not be considered further for publication in *eLife*.

The principle reasons for the decision were:

1) The reviewers were uncomfortable with the potential COI. The manuscript was viewed as not sufficiently differentiated from an advertisement. If the system design is to be open source, that should be clearly stated. If the system is to be sold commercially, then a clear COI statement must be included.

2) The system needs assessment of reproducibility across cohorts, validation with manual scoring and comparison to other existing systems.

If the authors can adequately address these issues, then *eLife* would be interested in taking another look.

*Reviewer #1:*

In general this is a valuable methodological contribution to the study of social interactions of mice. The system is novel and is likely to yield new data such as the VPA data provided as an example here.

Most of my suggestions are minor but a few overall comments are warranted.

The methods should be better integrated into the text to increase readability. Same with the supplementary figures which are beyond onerous. With the exception of code or circuit diagrams, put all figures into the manuscript in the order needed. Do not use supplemental figures.

What is the time limit for mice in the apparatus? 72 hr is used here – is that the limit and what is the limiting factor? How is air exchange handled (I am assuming that the system is closed on the top)?

How scalable is this system in terms of size? Can it be used for guinea pigs? rats?

The sociability measure is well described in the methods but the "Social preference" appears to me overly derived. Raw numbers – time in chamber with (and wo/) the social odor before during and after the odors presence would be better than the normalized factor used.

It is *very* confusing to use the term Social preference to refer to the mouse going to the mouse in a 3-chamber test and to a social ODOR in the eco-hab. I highly recommend using the term approach to social odor.

Throughout the manuscript, it is problematic that no non-social stimuli are tested – e.g. a non-social odor or an object. This limits the conclusions about the sociability per se of the measures. Nonetheless, the point here is a novel apparatus which could be used to make these direct comparisons even if they have not yet been made.

*Reviewer #2:*

This manuscript describes a four-chambered system for animal tracking using RFID tags and antennas as opposed to other available methods (video tracking, IR beam-breaks, piezoelectric floor, etc.) One major advantage of RFID tracking is that it allows for the testing of multiple animals in one arena with individual identifying information. The Eco-HAB apparatus described allows animals to explore a sizable space, the symmetry allows for comparison of different stimuli, and tube connectors may increase mouse comfort in the arena. The authors make an important argument about the need for better reproducibility in social testing, and provide data suggesting that social approach in anxiety-prone balb/c mice is different in the standard 3ChA test under high and low stress conditions. While they did not test the eco-HAB under different lighting or animal handling history conditions (making it not a direct comparison to either the 3ChA comparison or cross-lab variability), data from different time points suggests that the eco-HAB system provides consistent results. The authors further provide empirical data showing the utility of this system in assessing social behavioral differences across multiple variables (including mouse strain, VPA treatment, Fmr1 knockout). Finally, they present a neat metric they call "in-cohort sociability" that provides something really different from standard 3ChA testing because it measures social behavior towards known individuals. One might imagine a scenario under which novel animal social cue exploration might decrease but in-cohort sociability would go up, and having both metrics adds a lot to the interest of this setup.

1) This report may blur the boundary between a detailed description of a methodological advance and the advertisement of a commercial device (available for ~$2000 euros). Other *eLife* tools reports I've seen use commercial resources but not ones they are selling. I defer to the editor to determine suitability.

2) Data validation must be included so that the accuracy of the software and hardware are disclosed. In particular, a nontrivial amount of testing time should be dually scored by human video observers (e.g. at slow speed on visibly marked individuals) and the RFID data collection system. The authors mention that some signals are dropped, and this is a classic issue with RFID reads – fast moving animals can be missed, and partial tag reads can abound. Depending on the structure of the individual ID tags (unique at only the beginning, end, or all), partial tag reads may or may not be assignable to individuals. Comparisons to hand-scoring of video feed will allow the authors to report estimate how much missing data there is (as analyses reported consist of the subset of the reliable data after dropping conflicting signals. In one note they say that <1% of mice are read as far apart enough that they missed two antennas in a row, however this under-reports the frequency of error. There can be error with no discrepancy in the order of signals detected if, and mice can be missed while moving between adjacent chambers or staying for a long period of time in a single chamber where their entry and exit were both misread. A second method of scoring would put time values to this error (amount or% of time misclassified) as opposed to just frequency of signals being dropped from analysis.

3) This system could potentially provide the ability to do a tremendous amount with individual data, if this is supported in the analysis package. If this is supported (i.e. provided the pair-wise social behavior data in the manuscript were generated by the analysis package), the authors should discuss this potential. For example, a sub-group of cohoused mice could be treated and a subgroup could be untreated, and differences could be assessed within a single testing session. The heat maps of mouse pair interactions could be used to identify more and less social individual mice which could be useful for comparisons to other measures (for example individual differences in specific genes, neural markers, etc.). And presuming other small rodents like prairie or meadow voles could be tested in such an apparatus, stable affiliations could be examined between mates or same-sex peers. This represents an exciting advantage of this kind of tracking system that is currently overlooked in this presentation.

*Reviewer #2 (Additional data files and statistical comments):*

In my comments above I ask for the authors to share validation data with the results of two methods of scoring the same data. I would hope these have already been collected in the course of validating the hardware and software. It is painstaking work to validate 12 mice visually, but it needs to be done at least once for me to have any confidence in the output. I have seen multiple iterations of other similar testing systems have major flaws in the output that were not caught until such validation was performed.

*Reviewer #3:*

The authors propose a set-up to study social behavior in mice. The setup proposed is of a complexity between very simple standard tests and more complex RFID arenas. This intermediate level of complexity has the advantage of being much cheaper than full-area RFID arenas and the authors argue that the simplified arena is ethologically relevant.

While the authors compare their setup with simpler arenas, they don’t make a comparison with more complex arenas (Kimchi Lab). To know whether this simplified arena captures the necessary elements, it would seem necessary to compare against these full-area setups.

Perhaps one of the more relevant points is about reproducibility, but I see no discussion of the data apparently used to demonstrate it in Figure 5. What does it count as high reproducibility? Why the data would correspond to that standard? What are the bars in the graph?

The authors justify RFID approaches saying then video-based approaches have the problem of how to deal with shadows and corridors. For a fairer discussion, maybe the authors can (a) discuss how bad the problem really is (giving refs to new developments in this line) and (b) also discuss problems with RFID (more invasive than video?) and (c) try to discuss limitations of their proposed setup (cannot find in different lines how afraid to open spaces they are?).

*Reviewer #4:*

The paper entitled "Eco-HAB – fully automated and ecologically relevant assessment of social impairments in mouse models of autism" describes an innovative set-up to assess mouse social behavior in an automated way, without human intervention. Mouse sociability is assessed within the housing environment, that is, an arrangement of housing boxes connected by corridor tubes. Mice are tracked and localized using RFID technology. The data collected are used to estimate the amount of time each mouse spend within each box, with or without her conspecifics, and also to measure their interest for social odor cues in comparison with non-social odor cues. The authors challenged their system by testing several mouse models of autism. This new system should allow researchers to detect similar social impairments in different mouse models as the classic 3-chambered test. For example, mouse social interest could be analyzed in low-stress conditions. This could also spare time and avoid many confounding factors (such as experimenter biases, housing conditions, habituation to the test apparatus) and while this concept is of high interest for the community (replication is indeed a major issue in the field), I have several comments on this current version of the Eco-HAB.

First, this setting and the analyses conducted on the data are reduced to a very basic "social interaction" level. The social interaction is described as the time spent in which box with whom. This is disappointing given the high potential of the method to generate more precise data (e.g., sub-group formation: is it possible to quantify the number of individuals in the subgroup and how stable in time the subgroups are?). In addition, the supposition that a mouse spends time with another one when they are located in the same case is not clearly shown. The correction of the time spent with another mouse (by subtracting the supposed spontaneous (non-social) exploration to the time spent in the compartment) is not convincing. It would need further analyses and a validation. Indeed, on a few video samples, it might be possible to compare the manual scoring of time spent with another mouse to the amount of time calculated through the method presented. This would allow the authors to check the accuracy of the calculation. The non-social stimuli used in Eco-HAB might also be more elaborated. For instance, the authors use bedding with sent vs. fresh bedding as a test for social vs. non-social stimuli. To align with the 3-chambered test paradigm, for the non-social condition, the authors could use bedding + non-social odor such as lemon or bedding + inanimate object.

Second, the novelty of the system is not clear to me. The authors quoted a previous article by Weissbrod et al. published in Nature Communication entitled "Automated long-term tracking and social behavioral phenotyping of animal colonies within a semi-natural environment". The paper also describes a tracking via RFID and social interaction matrices. It would be important to compare the two systems and explicitly show why Eco-HAB is different or better. The authors should also include a table indicating the accuracy of the video-RFID-tracking system performance in their system (as Table1 in the Weissbrod paper).

Third, I don't think that "low cost custom system" and "reproducibility" is very relevant here. If the authors want to argue for the reproducibility of the results obtained with this system, it would be appropriate to indicate the results from the same measurements using different cohorts. Currently, the data of the different replications appeared to be pooled. In addition, reproducing the setup is relatively complex and expertise in electronics is mandatory to reproduce the system.

In summary, the authors did not convince the reviewer of the usefulness of this current version of Eco-HAB. However, if they present additional measures that can be made with Eco-HAB (and not with previous behavioral tests using RFID), the system could be of interest to increase reproducibility in the field of mouse behavior.

[Editors’ note: what now follows is the decision letter after the authors submitted for further consideration.]

Thank you for submitting your article "Eco-HAB as a fully automated and ecologically relevant assessment of social impairments in mouse models of autism" for consideration by *eLife*. Your article has been favorably evaluated by a Senior Editor and two reviewers, one of whom, Peggy Mason (Reviewer #1), is a member of our Board of Reviewing Editors. The following individual involved in the review of your submission has agreed to reveal their identity: Gonzalo de Polavieja (Reviewer #2).

The reviewers have discussed the reviews with one another and the Reviewing Editor has drafted this decision to help you prepare a revised submission.

This is a great revision. The remaining concerns are very specific and are listed below. Please take a look and revise accordingly. We look forward to seeing your revision.

*Reviewer #1:*

This is an important methodological advance in the field of rodent social behavior and neuroscience. The revised manuscript has addressed many of the concerns raised by the initial review. Of great importance, it is clear that the system is open source and this manuscript is a contribution and not an advertisement. Additional detail concerning reproducibility has been added.

Minor to moderate concerns are:

Please put open source into the Abstract and keywords and possibly also the summary.

Figure 1—figure supplement 2 – the legend is not helpful. What is being graphed on the y axis? It would appear to be crossings but that is not said in the legend. And how were the manual crossings measured – with red light? Most importantly, why is the interpretation that the RFID measurement is superior to the manual scoring? This graph shows a difference, a consistent one. But it is unclear to me which of the two is more accurate. Finally, this does not truly address the 3rd concern of the 4th reviewer which is admittedly a bit vague. But the gist of the concern is: how does the reader know that the two mice are "spending time together" when they both occupy the same box? I think this questions whether two mice in one box could be at opposite ends and facing away from each other and yet be counted as socially affiliating. Clearly a formal possibility. Do the authors have any data relevant to this concern?

The sentence “Even though, in the present form, our system does not allow for the recognition of particular littermate-related behaviors, results show that both Eco-HAB measures-in-cohort sociability and scent-based social approach allow for drawing similar conclusions” is not understandable to me.

The Discussion is short. Some text in the Results should be in the Discussion (e.g. subsection “Eco-HAB – ethologically relevant testing of social behaviors”, last two paragraphs; subsection “Eco-HAB measurement is unbiased by social hierarchy and allows for long-term monitoring of social behavior”, last paragraph). Several of the comments in the response to reviewers would also be useful to include. The manuscript is concise but to a fault. Hold the reader's hand a bit more and help them to understand what you know so well after working on this for years.

Figure 5. The original is in original form I believe and still does not speak to reproducibility as do Figure 5—figure supplements 1-3. The supplemental figures are very useful and should be the main figure. I do not know what Figure 5 shows in its present form. Here is a suggestion. Make different symbols for the two cohorts in each condition (wt vs. fmr1 and via vs. control). Then line them up from highest to lowest. As it is the individuals are ordered along the x axis by mouse number or some other arbitrary/meaningless parameter. This simply does not show replication. If anything it is a messy version of a scatter or box plot of the data showing variability and range.

What are the ovals in Figure 5—figure supplement 3? The dots are averages from the two days of testing?

*Reviewer #2:*

Most of the concerns I had have been appropriately address in the new version of the manuscript. Now it is clear that software and data are open. It is also more clear the comparison with other methods and how this setup is more reliable than others and that levels of stress are lower than in 3-chamber setups.

The comparison with standard 3-chamber setups is, in summary, quite deep. However, the comparison with open arenas (Weissbrod, 2103) is only verbal. The authors say that in open arenas animals have territorial fights, as recognized in Weissbrod 2013. However, I see no comparison now between the number of fights or stress levels in the present setup vs. Weissbrod 2013. The reader is left to assume that, because the setup is inspired in ethologically relevant behavior, it must obviously be true that there is less aggression. In the most beautiful scenario, we would have the wild, open arena and present setup measurements of aggressive encounters. In the next level at least a comparison between open arenas and this set-up. But, to avoid doing new experiments, I think the minimum would be to re-analize the data to measure aggression. Video data was acquired, so it should most probably be possible to analyze this data for aggression encounters.

---

## [Author Response]

[Editors’ note: the author responses to the first round of peer review follow.]

[…]

*1) The reviewers were uncomfortable with the potential COI. The manuscript was viewed as not sufficiently differentiated from an advertisement. If the system design is to be open source, that should be clearly stated. If the system is to be sold commercially, then a clear COI statement must be included.*

Thank you for pointing out this matter, as it gave us an opportunity to clearly state our intentions regarding Eco-HAB’s accessibility. The system has been designed to be open source both with respect to hardware and software. We have presented all the technical details and electronic solutions needed to construct the apparatus from scratch. We have also provided all the software required to run experiments and analyze data, along with sample implementations. Our only intention was to enable potentially interested scientists to build the apparatus on their own, without the necessity of purchasing any of its elements from us. The estimated cost of Eco-HAB was presented solely to stress the fact that it consists of inexpensive equipment, specifically that it was no more costly than traditional, non-automated set-ups serving similar purposes. As recommended, we have added a clear statement of Eco-HAB’s open source character in the manuscript (Introduction).

“[…]we designed Eco-HAB, a fully automated open source system, based on RFID technology and inspired by the results of ethological field studies in mice […]”.

*2) The system needs assessment of reproducibility across cohorts.*

In order to better demonstrate reproducibility, we have performed additional experiments as well as presented previously obtained data that was not part of the initial manuscript. To present them we designed an entirely new Figure 5 with 3 secondary graphics, addressing reproducibility between cohorts, reproducibility between results obtained in different laboratories and within subject comparisons.

*3) Validation with manual scoring.*

We added data on the efficiency of RFID antennas’ as compared to video-based, manual scoring. The implemented system of coils recognizes subjects with sensitivity far exceeding that obtained with standard video- and RFID-recognition methods. Implemented RFID coils are highly effective and have a sensitivity sufficient to register mice moving at their maximal speed. The RFID antennas run independently and do not disrupt each other’s workings, regardless of the signal’s amplitude. All coils may be activated simultaneously and register signals for unlimited time as described in the amended manuscript (Figure 1—figure supplement 2).The ability to simultaneously activate and run for an unlimited time is due to the synchronization system (output of a 4 MHz crystal oscillator powered with 5V distributed to all EM4095 IC clock inputs), which, as far as we know, has never been implemented in a laboratory animal recognition setup.

*4) And comparison to other existing systems.*

We added sentences to the manuscript (subsection “Eco-HAB – ethologically relevant testing of social behaviors”, last two paragraphs) stating how Eco-HAB compares to full-arena set-ups, such as the one proposed by Tali Kimchi’s Lab (Weissbrod A. et al. 2013). As shown in their publication, open-arena testing entails frequent display of aggressive behaviors, such as chasing or fighting. For studies of associative behaviors and stable affiliations among mice, we needed an environment that would reduce these types of interactions. It took 3 years of experimental trials and research into mouse ethology to come up with our system of tunnels and chambers that allows mice to display their natural affiliation patterns.

*If the authors can adequately address these issues, then eLife would be interested in taking another look.*

*Reviewer #1:*

*In general this is a valuable methodological contribution to the study of social interactions of mice. The system is novel and is likely to yield new data such as the VPA data provided as an example here.*

*Most of my suggestions are minor but a few overall comments are warranted.*

*The methods should be better integrated into the text to increase readability. Same with the supplementary figures which are beyond onerous. With the exception of code or circuit diagrams, put all figures into the manuscript in the order needed. Do not use supplemental figures.*

We designed the figures in accordance to *eLife* recommendations, which state that figures not central to the narrative should be presented as secondary figures, directly linked to the primary ones. We recognize that this kind of presentation may be troublesome for the reviewers, to whom all the figures are presented as separate files. However, potential readers will see all the graphics comprising a given figure and its supplements as comprehensive flash animations, inbuilt into the article’s body.

*What is the time limit for mice in the apparatus? 72 hr is used here – is that the limit and what is the limiting factor? How is air exchange handled (I am assuming that the system is closed on the top)?*

We made adjustments to the manuscript to clarify these matters: “Although the protocol utilized here lasted 72 h, it is noteworthy that, when required, Eco-HAB can run continuously even for months with only short technical breaks for cage cleaning (every 7 days). The cages are not connected to the IVC system, so air exchange is not an issue.”

*How scalable is this system in terms of size? Can it be used for guinea pigs? rats?*

The implemented solutions are easily scalable. With the exception of RFID coils, whose size needs to be adjusted in accordance to the corridor diagonal, there is no need to change applied electronics or software should one wish to adapt the system to species other than mice. This is stated in the amended manuscript (subsection “Eco-HAB - apparatus construction”).

*The sociability measure is well described in the methods but the "Social preference" appears to me overly derived. Raw numbers – time in chamber with (and wo/) the social odor before during and after the odors presence would be better than the normalized factor used.*

When animals are tested in Eco-HAB or using manual behavioral tests, they usually display a preference for one of the experimental compartments during the adaptation phase. We argue that this factor should be taken under consideration, as it may influence subsequent measurement of social preference. We consider this issue to be even more prominent in automated set-ups, where animals are exposed to the experimental environment 24h a day. Since Eco-HAB allows testing of spontaneous behavior in animals, their habits might play an even greater role. Additionally, animals’ activity, which usually decreases in the course of experiment, may confound interpretation based solely on raw data. For these reasons we think that the normalized factor we present is the most accurate way to assess social preference.

*It is very confusing to use the term Social preference to refer to the mouse's going to the mouse in a 3-chamber test and to a social ODOR in the eco-hab. I highly recommend using the term approach to social odor.*

The text, figures and their descriptions have been adjusted according to your suggestion.

*Throughout the manuscript, it is problematic that no non-social stimuli are tested – e.g. a non-social odor or an object. This limits the conclusions about the sociability per se of the measures. Nonetheless, the point here is a novel apparatus which could be used to make these direct comparisons even if they have not yet been made.*

We fully agree that when testing social preference one should introduce non-social stimulus for control purposes. That is why, as described in Materials and methods section (see: Three-chambered apparatus testing, Eco-HAB testing), in all of our experiments we use non-social stimuli (a non-social object or a non-social odor respectively) in order to control for novelty – a factor that might cause an increase in exploratory behaviors, such as sniffing and approaching. We also compared responses to social and non-social stimulus of the same modality.

Reviewer #2: […]

*1) This report may blur the boundary between a detailed description of a methodological advance and the advertisement of a commercial device (available for ~$2000 euros). Other eLife tools reports I've seen use commercial resources but not ones they are selling. I defer to the editor to determine suitability.*

The system has been designed to be open source in the full meaning of the word. We presented all the technical details and electronic details needed to construct the apparatus from scratch. We also provided all the software required to run experiments and analyze data, along with example implementations. Our only intention was to enable potentially interested scientist to build the apparatus on their own, without the necessity of purchasing any of its elements from us. The estimated cost of the Eco-HAB was presented solely to stress the fact that it was inexpensive equipment, specifically that it was no more costly than traditional, non-automated behavioral set-ups serving similar purposes. We added a clear statement of Eco-HAB’s open source character in the manuscript (Introduction, sixth paragraph).

*2) Data validation must be included so that the accuracy of the software and hardware are disclosed. In particular, a nontrivial amount of testing time should be dually scored by human video observers (e.g. at slow speed on visibly marked individuals) and the RFID data collection system. The authors mention that some signals are dropped, and this is a classic issue with RFID reads – fast moving animals can be missed, and partial tag reads can abound. Depending on the structure of the individual ID tags (unique at only the beginning, end, or all), partial tag reads may or may not be assignable to individuals. Comparisons to hand-scoring of video feed will allow the authors to report estimate how much missing data there is (as analyses reported consist of the subset of the reliable data after dropping conflicting signals. In one note they say that <1% of mice are read as far apart enough that they missed two antennas in a row, however this under-reports the frequency of error. There can be error with no discrepancy in the order of signals detected if, and mice can be missed while moving between adjacent chambers or staying for a long period of time in a single chamber where their entry and exit were both misread. A second method of scoring would put time values to this error (amount or% of time misclassified) as opposed to just frequency of signals being dropped from analysis.*

We have added data on the sensitivity of the RFID system we had obtained before we started to perform experiments. The data includes comparisons of the efficiency of the antennas with respect to video-based manual scoring (Figure 1—figure supplement 2). The system of coils (subsection “Eco-HAB - applied electronic solutions”) can recognize subjects with high sensitivity, far exceeding that obtained with standard video- and RFID-recognition methods. These RFID coils have sufficient sensitivity to register even mice moving at high speed. This is all possible because of the internal 4 MHz crystal oscillator synchronization. To the best of our knowledge, this has never been implemented in laboratory animal recognition set-ups. The RFID antennas run independently and do not disrupt each other’s workings, regardless of signal’s amplitude. Moreover, all coils may be active simultaneously and register signals for practically unlimited time. Since the system is designed to be highly sensitive in order not to miss animals’ signals when they are running, it is expected that it would generate superfluous readouts (e.g. when animal is sitting under antenna). If present, such readouts are later eliminated by the Eco-HAB.py software, that contains algorithms recognizing such events. We have also described (Materials and methods, section Eco-HAB.py – software package for data processing and analysis: Data processing algorithm) how our software deals with cases when signal was dropped. Even if this happens, it would not influence the final assessment of social preference. As to partial tag readouts, they are not an issue since such events are absent in our system.

*3) This system could potentially provide the ability to do a tremendous amount with individual data, if this is supported in the analysis package. If this is supported (i.e. provided the pair-wise social behavior data in the manuscript were generated by the analysis package), the authors should discuss this potential. For example, a sub-group of cohoused mice could be treated and a subgroup could be untreated, and differences could be assessed within a single testing session. The heat maps of mouse pair interactions could be used to identify more and less social individual mice which could be useful for comparisons to other measures (for example individual differences in specific genes, neural markers, etc.). And presuming other small rodents like prairie or meadow voles could be tested in such an apparatus, stable affiliations could be examined between mates or same-sex peers. This represents an exciting advantage of this kind of tracking system that is currently overlooked in this presentation.*

Thank you for this feedback. Indeed – Eco-HAB can provide all the features you have mentioned and we had not stressed this enough in the initial manuscript. All the individual data can easily be analyzed using our software package. The software is open source code, enabling users to modify it to develop their own custom measures. Analysis of pair-wise social behavior we reported is one of the default functionalities of Eco-HAB.py. Closely following your advice we have modified the Results section (see: Eco-HAB – ethologically relevant testing of social behaviors) with text regarding Eco-HAB’s potential to analyze individual social behavior in a detailed manner as well as track stable affiliations and test littermate-interactions of sub-groups of mice within one experimental session: “Besides asserting described measures of ethologically relevant sociability, Eco-HAB provides the possibility of performing in-depth analysis of individual social behaviors. […] Compared to testing experimental and control groups separately, such a solution allows for evaluation of social environment as an essential factor that may influence littermate-related behavior.”’

*Reviewer #2 (Additional data files and statistical comments):*

*In my comments above I ask for the authors to share validation data with the results of two methods of scoring the same data. I would hope these have already been collected in the course of validating the hardware and software. It is painstaking work to validate 12 mice visually, but it needs to be done at least once for me to have any confidence in the output. I have seen multiple iterations of other similar testing systems have major flaws in the output that were not caught until such validation was performed.*

As described above, we had already performed such an analysis before the system began to be utilized for experimental purposes. As we stated, the RFID coils are highly effective and have sensitivity sufficient to register running mice. We included this data and a description of how it was obtained in the manuscript (subsection “Eco-HAB - applied electronic solutions**”**, Figure 1—figure supplement 2).

*Reviewer #3:*

*The authors propose a set-up to study social behavior in mice. The setup proposed is of a complexity between very simple standard tests and more complex RFID arenas. This intermediate level of complexity has the advantage of being much cheaper than full-area RFID arenas and the authors argue that the simplified arena is ethologically relevant.*

*While the authors compare their setup with simpler arenas, they don’t make a comparison with more complex arenas (Kimchi Lab). To know whether this simplified arena captures the necessary elements, it would seem necessary to compare against these full-area setups.*

We have added information to the manuscript (subsection “Eco-HAB – ethologically relevant testing of social behaviors”, last two paragraphs) stating how Eco-HAB compares to full-arena set-ups, such as that proposed by Tali Kimchi’s Lab (Weissbrod A. et al. 2013). Basically, our system avoids the extensive territorial fights that occur in open arena settings. This is very important since our goal is to measure associative behaviors (stable affiliations) among mice of the same as well as opposite sexes. As shown by Weissbrod et al. (2013) open-arena testing entails frequent display of aggressive behaviors, such as chasing. We consider feasibility of observing spontaneous, long lasting affiliations between conspecifics an exciting advantage of Eco-HAB as a tracking system. Additional design elements were inspired by the work of Weissbrod et al. (2013) including our technical RFID electronics solutions. However, in Eco-HAB, transponders do not have to be perpendicular to the antennas in order to get optimal signal strength. This results in superior registration accuracy which is independent of the position of an animal. Additionally, we placed two antennas on each corridor leading to the housing compartments to be able to explicitly assess the direction from which animal was entering.

*Perhaps one of the more relevant points is about reproducibility, but I see no discussion of the data apparently used to demonstrate it in Figure 5. What does it count as high reproducibility? Why the data would correspond to that standard? What are the bars in the graph?*

In order to make the reproducibility issue more convincing, we have performed additional experiments, as well as presented previously obtained data, that had not been a part of the initial manuscript. To do so, we constructed an entirely new Figure 5 with 3 secondary graphics, addressing reproducibility between cohorts, reproducibility between results obtained in different laboratories and within subject comparisons.

*The authors justify RFID approaches saying then video-based approaches have the problem of how to deal with shadows and corridors. For a fairer discussion, maybe the authors can (a) discuss how bad the problem really is (giving refs to new developments in this line) and (b) also discuss problems with RFID (more invasive than video?) and (c) try to discuss limitations of their proposed setup (cannot find in different lines how afraid to open spaces they are?).*

Following your advice we added comments concerning how our system is an improvement on video-based methods, the invasive character of RFID-tracking, and limitations of Eco-HAB system, which in the present form, does not allow for the recognition of particular littermate-related behaviors (Introduction, fifth paragraph and subsection “Eco-HAB – ethologically relevant testing of social behaviors”, first paragraph).

*Reviewer #4:*

*The paper entitled "Eco-HAB – fully automated and ecologically relevant assessment of social impairments in mouse models of autism" describes an innovative set-up to assess mouse social behavior in an automated way, without human intervention. Mouse sociability is assessed within the housing environment, that is, an arrangement of housing boxes connected by corridor tubes. Mice are tracked and localized using RFID technology. The data collected are used to estimate the amount of time each mouse spend within each box, with or without her conspecifics, and also to measure their interest for social odor cues in comparison with non-social odor cues. The authors challenged their system by testing several mouse models of autism. This new system should allow researchers to detect similar social impairments in different mouse models as the classic 3-chambered test. For example, mouse social interest could be analyzed in low-stress conditions. This could also spare time and avoid many confounding factors (such as experimenter biases, housing conditions, habituation to the test apparatus) and while this concept is of high interest for the community (replication is indeed a major issue in the field), I have several comments on this current version of the Eco-HAB.*

*First, this setting and the analyses conducted on the data are reduced to a very basic "social interaction" level. The social interaction is described as the time spent in which box with whom. This is disappointing given the high potential of the method to generate more precise data (e.g., sub-group formation: is it possible to quantify the number of individuals in the subgroup and how stable in time the subgroups are?).*

Eco-HAB can provide the features you have mentioned, namely results on sub-group formation and stability, although perhaps we had not stressed this issue enough in the manuscript. All of the individual and sub-group data can easily be analyzed through our software package. The software is an open source code which means that users can modify it to obtain custom measures. Analysis of pair-wise social behavior is one of the default functionalities of Eco-HAB.py. Following your advice we added comments to the Results and Discussion section (see: Eco-HAB – ethologically relevant testing of social behaviors) regarding Eco-HAB’s potential to analyze individual social behavior as well as track stable affiliations and test littermate-interactions of sub-groups of mice within one experimental session (–subsection “Eco-HAB – ethologically relevant testing of social behaviors”, last two paragraphs).

*In addition, the supposition that a mouse spends time with another one when they are located in the same case is not clearly shown. The correction of the time spent with another mouse (by subtracting the supposed spontaneous (non-social) exploration to the time spent in the compartment) is not convincing. It would need further analyses and a validation. Indeed, on a few video samples, it might be possible to compare the manual scoring of time spent with another mouse to the amount of time calculated through the method presented. This would allow the authors to check the accuracy of the calculation. The non-social stimuli used in Eco-HAB might also be more elaborated. For instance, the authors use bedding with sent vs. fresh bedding as a test for social vs. non-social stimuli. To align with the 3-chambered test paradigm, for the non-social condition, the authors could use bedding + non-social odor such as lemon or bedding + inanimate object.*

Before we started to perform animal experiments in Eco-HAB, we had already completed a detailed analysis of the RFID system efficiency. This was done by means of comparison with video recordings. We describe this in detail in addressing your comment regarding RFID antennas (see below). In accordance to your advice, we have added new charts (Figure 1—figure supplement 3) concerning system validation.

As for the olfactory stimuli utilized in Eco-HAB, in these experiments we deliberately presented social and non-social stimuli of the same modality in order to appropriately control for novelty. Moreover, we avoided using stimuli that might be associated with food or sex in order to eliminate the possibility of motivational conflicts that might have confounded assessment of sociability. We have made alterations to the manuscript explicitly stating the possibility of being able to use any scent (subsection “Eco-HAB testing”, last paragraph) depending on the particular scientific question being asked.

*Second, the novelty of the system is not clear to me. The authors quoted a previous article by Weissbrod et al. published in Nature Communication entitled "Automated long-term tracking and social behavioral phenotyping of animal colonies within a semi-natural environment". The paper also describes a tracking via RFID and social interaction matrices. It would be important to compare the two systems and explicitly show why Eco-HAB is different or better. The authors should also include a table indicating the accuracy of the video-RFID-tracking system performance in their system (as Table1 in the Weissbrod paper).*

We have added information to the manuscript (subsection “Eco-HAB – ethologically relevant testing of social behaviors”, last two paragraphs) stating how Eco-HAB compares to full-arena set-ups, such as that proposed by Tali Kimchi’s Lab (Weissbrod A. et al. 2013). Basically, our system avoids the extensive territorial fights that occur in open arena settings. This is very important since our goal is to measure associative behaviors (stable affiliations) among mice of the same as well as opposite sexes. As shown by Weissbrod et al. (2013) open-arena testing entails frequent display of aggressive behaviors, such as chasing. We consider feasibility of observing spontaneous, long lasting affiliations between conspecifics an exciting advantage of Eco-HAB as a tracking system. Additional design elements were inspired by the work of Weissbrod et al. (2013) including our technical RFID electronics solutions. However, In Eco-HAB, transponders do not have to be perpendicular to the antennas in order to get optimal signal strength. This results in superior registration accuracy which is independent of the position of an animal. Additionally, we placed two antennas on each corridor leading to the housing compartments to be able to explicitly assess the direction from which animal was entering.

We have also added data on the sensitivity of the RFID system we had obtained before we started to perform experiments. The data includes comparisons of the efficiency of the antennas with respect to video-based manual scoring (Figure 1—figure supplement 2). The system of coils (subsection “Eco-HAB - applied electronic solutions”) can recognize subjects with high sensitivity, far exceeding that obtained with standard video- and RFID-recognition methods. These RFID coils have sufficient sensitivity to register even mice moving at high speed. This is all possible because of the internal 4 MHz crystal oscillator synchronization. To the best of our knowledge, this has never been implemented in laboratory animal recognition set-ups. The RFID antennas run independently and do not disrupt each other’s workings, regardless of signal’s amplitude. Moreover, all coils may be active simultaneously and register signals for practically unlimited time. Since the system is designed to be highly sensitive in order not to miss animals’ signals when they are running, it is expected that it would generate superfluous readouts (e.g. when animal is sitting under antenna). If present, such readouts are later eliminated by the Eco-HAB.py software, that contains algorithms recognizing such events. We have also described (Materials and methods, section Eco-HAB.py – software package for data processing and analysis: Data processing algorithm) how our software deals with cases when signal was dropped. Even if this happens, it would not influence the final assessment of social preference. As to partial tag readouts, they are not an issue since such events are absent in our system.

*Third, I don't think that "low cost custom system" and "reproducibility" is very relevant here. If the authors want to argue for the reproducibility of the results obtained with this system, it would be appropriate to indicate the results from the same measurements using different cohorts. Currently, the data of the different replications appeared to be pooled. In addition, reproducing the setup is relatively complex and expertise in electronics is mandatory to reproduce the system.*

In order to make the reproducibility issue more convincing, we have performed additional experiments, as well as presented previously obtained data, that had not been a part of the initial manuscript. To do so, we constructed an entirely new Figure 5 with 3 secondary graphics, addressing reproducibility between cohorts, reproducibility between results obtained in different laboratories and within subject comparisons. As recommended we have presented results from the same measurements using different cohorts (Figure 5).

As to the electronic, we have presented all the data needed for reproduction of a new Eco-HAB setup. Even if a scientist has insufficient electronic expertise to build one themselves, they could order a system at low cost from someone who did. All electronic elements are inexpensive and easily obtainable. Design of the system, which would be the most expensive part, has already been taken care of by us. Moreover, we would be happy to assist any potentially interested laboratories with implementation of their own systems.

*In summary, the authors did not convince the reviewer of the usefulness of this current version of Eco-HAB. However, if they present additional measures that can be made with Eco-HAB (and not with previous behavioral tests using RFID), the system could be of interest to increase reproducibility in the field of mouse behavior.*

As described above (subsection “Eco-HAB – ethologically relevant testing of social behaviors”, last two paragraphs), we strongly believe that the ability to measure spontaneous, long lasting affiliations between mice is the greatest advantage of Eco-HAB as compared to previously published set-ups. This is achievable only because we were able to minimize aggressive and territorial behaviors, that are intrinsic to commonly applied open-arena testing. To our knowledge, Eco-HAB is a first system enabling animals to inhabit multiple small sub-territories, resembling natural burrows. We argue, that this is due to the lack of attractive resources placed in the middle of the area available for exploration. We hope that, together with the data showing the reproducibility of Eco-HAB measurements we were able to convince you of its usefulness, especially for studies concerning non-confrontational littermate-related behaviors in rodents.

[Editors' note: the author responses to the re-review follow.]

*Reviewer #1:*

*This is an important methodological advance in the field of rodent social behavior and neuroscience. The revised manuscript has addressed many of the concerns raised by the initial review. Of great importance, it is clear that the system is open source and this manuscript is a contribution and not an advertisement. Additional detail concerning reproducibility has been added.*

*Minor to moderate concerns are:*

*Please put open source into the Abstract and keywords and possibly also the summary.*

We made appropriate changes in the manuscript (Abstract, Keywords and Conclusion, first paragraph).

*Figure 1—figure supplement 2 – the legend is not helpful. What is being graphed on the y axis? It would appear to be crossings but that is not said in the legend. And how were the manual crossings measured – with red light? Most importantly, why is the interpretation that the RFID measurement is superior to the manual scoring? This graph shows a difference, a consistent one. But it is unclear to me which of the two is more accurate.*

We have designed a new Figure 1—figure supplement 2 with altered legend. The implemented RFID antennas are far more sensitive than what is needed to recognize every visually registered crossing. Now the graph shows that for every video-recorded event (animal passing under the antenna) there are on average 3.9/3.4 (2 independent measurements) RFID registrations. Superfluous data are automatically screened and removed by the Eco-HAB.py software, as described in the subsection “Data processing algorithm”. Further, as described in subsection “Eco-HAB – applied electronic solutions” video-based assessment of antenna crossings was performed during the first 6 hour period of the dark phase at the beginning of the adaptation phase, when animals are most active and intensively explore a new environment (subsection “Eco-HAB - applied electronic solutions”, third paragraph). As advised, we explicitly stated the conditions under which videos were recorded, namely that they were obtained in complete darkness. The source of infra-red light placed above the apparatus was used to illuminate the field of view (in the aforementioned paragraph).

*Finally, this does not truly address the 3rd concern of the 4th reviewer which is admittedly a bit vague. But the gist of the concern is: how does the reader know that the two mice are "spending time together" when they both occupy the same box? I think this questions whether two mice in one box could be at opposite ends and facing away from each other and yet be counted as socially affiliating. Clearly a formal possibility. Do the authors have any data relevant to this concern?*

We added to the Discussion the passage that draws attention of a reader to such possibility: “Nevertheless, in the present form, our system does not allow for the recognition of particular types of subtle littermate-related behaviors that might skew results such as having two animals in the same chamber but facing away from each other and not interacting. While it is not possible to distinguish different types of social interactions yet, casual observations of video recordings obtained during numerous experiments lead us to believe that such events are rather accidental.”

*The sentence “Even though, in the present form, our system does not allow for the recognition of particular littermate-related behaviors, results show that both Eco-HAB measures-in-cohort sociability and scent-based social approach allow for drawing similar conclusions” is not understandable to me.*

The sentence was modified: “Nevertheless, in the present form, our system does not allow for the recognition of particular types of subtle littermate-related behaviors (…)”.

*The Discussion is short. Some text in the Results should be in the Discussion (e.g. subsection “Eco-HAB – ethologically relevant testing of social behaviors “, last two paragraphs; subsection “Eco-HAB measurement is unbiased by social hierarchy and allows for long-term monitoring of social behavior”, last paragraph). Several of the comments in the response to reviewers would also be useful to include. The manuscript is concise but to a fault. Hold the reader's hand a bit more and help them to understand what you know so well after working on this for years.*

The Discussion section was enriched and modified. Please find the changes we implemented below.

“Unlike short-term assessment using manual tests, Eco-HAB allows for long-term monitoring of social behaviors. Importantly, data collected in Eco-HAB show that the dynamics of response to social stimuli may differ depending on the tested strain of mice (see Figure 3—figure supplement 2).”

“In contrast to available open-arena set-ups, the apparatus we constructed allows mice to display their natural affiliation patterns. […] One can also envision expanding these measurements beyond mice to include other rodents, such as prairie or meadow voles.”

“A noteworthy asset of the Eco-HAB apparatus is the free, custom software which aids in obtaining effective measurements and speeds up data analysis. Appropriate programs were created for the purpose of data collection and conversion as well as in-depth evaluation of social behaviors. This code is open source and can be expanded to encompass new analyses. Assessing reliability of different behavioral measures, we chose the most valid.”

“Nevertheless, in the present form, our system does not allow for the recognition of particular types of subtle littermate-related behaviors that might skew results such as having two animals in the same chamber but facing away from each other and not interacting. While it is not possible to distinguish different types of social interactions yet, casual observations of video recordings obtained during numerous experiments lead us to believe that such events are rather accidental.”

*Figure 5. The original is in original form I believe and still does not speak to reproducibility as do Figure 5—figure supplements 1-3. The supplemental figures are very useful and should be the main figure. I do not know what Figure 5 shows in its present form. Here is a suggestion. Make different symbols for the two cohorts in each condition (wt vs. fmr1 and via vs. control). Then line them up from highest to lowest. As it is the individuals are ordered along the x axis by mouse number or some other arbitrary/meaningless parameter. This simply does not show replication. If anything it is a messy version of a scatter or box plot of the data showing variability and range.*

Figure 5 and its legend: “(…) Each column represents one cohort of animals, while data points (dots and squares) represent scores of particular mice.”) was adjusted in accordance to your suggestions, to clearly show scores of mice grouped in a given cohort. Further, as advised we also changed Figure 5—figure supplements 1-3 into main figures numbered 6 to 8.

*What are the ovals in Figure 5—figure supplement 3? The dots are averages from the two days of testing?*

Figure 5—figure supplement 3 (now Figure 8) has been described more precisely. We have adjusted the legend: “Eco-HAB allows remarkably reproducible assessment of approach to social odor in both (a) wild-type mice (n=9) and (b) Fmr1 knockouts (n=11). Evaluation of social behavior of subjects was repeated twice in identical Eco-HAB experiments, separated by a 10-day period of regular housing. Each aligned dot and square encircled by an oval represent individual score of approach to social odor for each tested mouse, measured in two subsequent experimental repetitions. Dots are data, while the ovals serve to guide the eye. Data presented are logarithmic values.”

*Reviewer #2:*

*Most of the concerns I had have been appropriately address in the new version of the manuscript. Now it is clear that software and data are open. It is also more clear the comparison with other methods and how this setup is more reliable than others and that levels of stress are lower than in 3-chamber setups.*

*The comparison with standard 3-chamber setups is, in summary, quite deep. However, the comparison with open arenas (Weissbrod, 2103) is only verbal. The authors say that in open arenas animals have territorial fights, as recognized in Weissbrod 2013. However, I see no comparison now between the number of fights or stress levels in the present setup vs. Weissbrod 2013. The reader is left to assume that, because the setup is inspired in ethologically relevant behavior, it must obviously be true that there is less aggression. In the most beautiful scenario, we would have the wild, open arena and present setup measurements of aggressive encounters. In the next level at least a comparison between open arenas and this set-up. But, to avoid doing new experiments, I think the minimum would be to re-analize the data to measure aggression. Video data was acquired, so it should most probably be possible to analyze this data for aggression encounters.*

Following your advice we quantified number and time of aggressive encounters for every tested strain and presented them in the new figure (see Figure 9—figure supplement 1 and its legend: “Figure 9—figure supplement 1. Aggressive interactions during testing in Eco-HAB are rare regardless of the tested strain. Number of episodes and duration of aggressive behaviors in VPA-treated and control BALB/c (a, b), VPA-treated and control C57BL/6 (c, d) and Fmr1 knockout and wild-type mice (e, f) during first 6 hours of adaptation phase, as counted per each pair of animals within a tested cohort. Aggressive encounters, namely fighting, chasing and biting were quantified by manual video-based scoring and then divided by the number of mouse pairs in a given cohort.”). Behaviors were measured by manual video-based scoring of previously obtained recordings. We assessed aggression encounters during first 6 hour period of the dark phase at the beginning of the adaptation period, the time when animals are most active, intensively exploring new environment and when potential aggressive behaviors were most probable, due to yet unstable social relations on novel territory (description added in Materials and methods section, see subsection “Measurement of aggressive encounters in Eco-HAB”).